# Clonal origin and development of high hyperdiploidy in childhood acute lymphoblastic leukaemia

Eleanor L. Woodward[1,9], Minjun Yang [1,9], Larissa H. Moura-Castro [1], Hilda van den Bos[2], Rebeqa Gunnarsson[1], Linda Olsson-Arvidsson[1,3], Diana C. J. Spierings [2], Anders Castor[4], Nicolas Duployez[5,6], Marketa Zaliova [7,8], Jan Zuna [7,8], Bertil Johansson [1,3], Floris Foijer [2] & Kajsa Paulsson [1] ✉

High hyperdiploid acute lymphoblastic leukemia (HeH ALL), one of the most common childhood malignancies, is driven by nonrandom aneuploidy (abnormal chromosome numbers) mainly comprising chromosomal gains. In this study, we investigate how aneuploidy in HeH ALL arises. Single cell whole genome sequencing of 2847 cells from nine primary cases and one normal bone marrow reveals that HeH ALL generally display low chromosomal heterogeneity, indicating that they are not characterized by chromosomal instability and showing that aneuploidy-driven malignancies are not necessarily chromosomally heterogeneous. Furthermore, most chromosomal gains are present in all leukemic cells, suggesting that they arose early during leukemogenesis. Copy number data from 577 primary cases reveals selective pressures that were used for in silico modeling of aneuploidy development. This shows that the aneuploidy in HeH ALL likely arises by an initial tripolar mitosis in a diploid cell followed by clonal evolution, in line with a punctuated evolution model.

The genetic origin of tumours remains obscure as the earliest stages of tumorigenesis cannot be observed. In the classic view of tumour development, cells acquire mutations in a stepwise manner, with clonal selection shaping the tumour genome over time and genomic heterogeneity arising by branching of different subclones[1,2]. However, in recent years this view has been challenged by data showing that some tumours arise by punctuated evolution, where the bulk of genetic aberrations occur within a short time frame at tumour initiation, followed by proliferation during which only little additional genomic heterogeneity is added[1,3,4]. The punctuated evolution model appears to fit particularly well with copy number aberrations, both intrachromosomal and those involving whole chromosomes[1].

The high hyperdiploid (HeH; 51-67 chromosomes) subtype comprises 25–30% of all paediatric B-cell precursor acute lymphoblastic leukaemia (ALL). HeH ALL is characterized by nonrandom chromosomal gains predominately involving 1–2 extra copies of chromosomes X, 4, 6, 10, 14, 17, 18, and 21, whereas chromosomal losses are very rare[5]. Several lines of evidence suggest that the aneuploidy arises early in

[1]Department of Laboratory Medicine, Division of Clinical Genetics, Lund University, Lund, Sweden. [2]European Research Institute for the Biology of Ageing (ERIBA), University of Groningen, University Medical Center Groningen, Groningen, The Netherlands. [3]Department of Clinical Genetics, Pathology, and Molecular Diagnostics, Office for Medical Services, Region Skåne, Lund, Sweden. [4]Department of Pediatrics, Skåne University Hospital, Lund University, Lund, Sweden. [5]Laboratory of Hematology, Centre Hospitalier Universitaire (CHU) Lille, Lille, France. [6]Unité Mixte de Recherche en Santé (UMR-S) 1172, INSERM/University of Lille, Lille, France. [7]Department of Pediatric Hematology and Oncology, Second Faculty of Medicine, Charles University/University Hospital Motol, Prague, Czech Republic. [8]Childhood Leukaemia Investigation Prague (CLIP), Prague, Czech Republic. [9]These authors contributed equally: E. L. Woodward, M. Yang. ✉e-mail: kajsa.paulsson@med.lu.se

HeH ALL, possibly already before birth[6–10], although overt leukaemia does not occur until several years later. Furthermore, analyses of allelic ratios in tetrasomic chromosomes have suggested that the extra chromosomes are gained at the same time in one abnormal cell division[11–13]. However, the details on how HeH ALL develops genetically remain unknown.

We have addressed the origin of HeH ALL using single cell whole genome sequencing (scWGS), analyses of selection pressures in a large patient cohort, and through in silico modelling. We find that stable aneuploid karyotypes that we observe in HeH ALL likely arise during a single tripolar mitosis followed by low-level clonal evolution. Our findings shed light into the earliest stages of tumorigenesis of the most common malignancy in childhood.

## Results

### HeH ALL displays little genomic heterogeneity

To understand how the aneuploidy arises in HeH ALL, we first set out to determine the degree of genomic heterogeneity, in particular chromosomal heterogeneity as a readout of chromosomal instability (CIN). We performed low-pass scWGS of 257–348 individual bone marrow cells/case, in total 2847 cells, from nine primary hyperdiploid ALL cases (2–13 years old at diagnosis; median 5 years) and one normal bone marrow sample (Supplementary Table 1). Copy number analysis for each individual cell was carried out with a resolution of approximately 5 Mb. For some chromosomes, we investigated which chromosomal homologue that was gained, lost, or displayed uniparental isodisomy (UPID; disomies involving two copies of the same chromosomal homologue) taking advantage of heterozygous variants identified through bulk WGS of matched samples. Phylogenetic trees were then constructed based on the combined data from scWGS, bulk WGS, and fluorescence in situ hybridization (FISH).

The normal bone marrow displayed diploidy in 269/270 cells (99.6%), with only one cell deviating by loss of chromosome 21, showing the high quality of the scWGS (Fig. 1). Of the 2577 cells in the leukaemic samples, five were normal diploid cells and the rest showed copy number changes agreeing with leukaemic cells. Overall, highly homogeneous genomes were seen for most of the leukaemias (Fig. 1), with predominantly whole chromosome gains being present in all cells. When assessing whole chromosome changes, 5/9 cases had the same chromosomal content in >99% of the cells, with only 1–2 cells displaying gains or losses of single chromosomes that were not seen in the other cells, suggesting a chromosome missegregation rate highly similar as observed for the normal bone marrow. The remaining four cases had 3–5 numerical subclones each (a clone being defined as at least two cells with the same genetic aberrations), with the major clone making up 55–88% of the cells (Table 1). For 3/4 cases, at least one of these subclones was also detectable in copy number analysis of bulk DNA; i.e. they would appear to harbour subclones also by this method. Case 2, however, displayed three minor subclones, each corresponding to 2.7–3.9% of the cells, which analysis of bulk DNA failed to detect so that it appeared to have only one clone. Analysis of chromosomal homologues revealed hidden heterogeneity in #9, where trisomy 17 involved different homologues in two distinct cell populations (Supplementary Fig. 1); this was, however, the only case of haplotype heterogeneity found among the 62 chromosomal gains/UPIDs that could be investigated.

Next, we calculated heterogeneity scores for each case (Table 1). There was no correlation between the heterogeneity scores and the number of cells sequenced, showing that the results were not skewed based on the number of cells included ($r_s = -0.38$, $P = 0.32$; two-sided Spearman's correlation test; Supplementary Fig. 2). Cases with relatively few subclones (#1, #5, #6, #7, and #8) had lower scores than cases with more subclones (#2, #3, #4, and #9). To investigate whether the observed differences in heterogeneity were due to mutations in genes affecting genomic stability, we screened bulk WGS data, but no

such correlation was seen (Supplementary Table 1). We further investigated whether increased heterogeneity correlated with the presence of sister chromatid cohesion defects in metaphase chromosomes, which we have recently reported to be associated with increased chromosomal heterogeneity in HeH ALL[14]. Indeed, #2, which had the second highest heterogeneity score, had a very high frequency of cohesion defects in metaphase cells (85%; Table 1). However, #3, #4 and #9, which also had high heterogeneity scores, had relatively few cells with cohesion defects. Overall, however, although the heterogeneity scores varied between cases, all had relatively low levels of heterogeneity, with non-clonal changes only seen in 0–2.6% of the cells.

In conclusion, the scWGS analysis revealed very low to low chromosomal heterogeneity in HeH childhood ALL. Thus, these leukaemias appear to have relatively stable genomes, despite being aneuploid.

### The chromosomal gains are early and ubiquitously present in HeH ALL

To understand how hyperdiploidy develops in the absence of CIN, we studied the phylogenetic trees of the chromosomal changes (Fig. 2). In all cases, the majority of chromosomal gains were seen at the roots of the trees, with most remaining stable and unchanging. The pattern of chromosomal gains in the inferred initial leukaemic cells resembled the one usually seen in HeH ALL: chromosomes X (100%), 21 (100%), 4 (89%), 14 (89%), 18 (89%), 6 (67%), 10 (67%), 17 (67%), 8 (44%), 9 (33%), 5 (22%), 16 (22%), 3 (11%), 11 (11%), and 12 (11%). Looking at chromosomal gains only, those in the earliest clone and in the major clone were identical in 5/9 (56%) of the cases, with the remaining four cases differing by gain or loss of 1–2 chromosomes. Thus, most extra chromosomes found at diagnosis were acquired early in leukemogenesis, in line with previous studies of HeH ALL[6–10]. Calculation of phylogenetic distances showed long truncal and short branching distances, suggesting punctuated evolution (Supplementary Fig. 3)[1]. Chromosomes that changed in copy number during clonal evolution comprised X, 8, 9, 14, 16, 17, and 21 (Fig. 2). Several cases displayed more than one instance of a particular chromosomal copy number change during their clonal evolution, comprising losses of 9 (two events in #2), gains of 17 (two events in #9), and gains of 21 (two events in #4), indicating strong clonal selection for these changes.

Only few clonal structural changes leading to copy number changes were detected by scWGS, in line with such events being relatively rare in hyperdiploid ALL[10]. In 7/9 cases, no structural changes were present in the inferred earliest cell, indicating that such abnormalities typically arose after the bulk of the chromosomal gains (Fig. 2). Duplication of 1q [dup(1q)] was seen in subclones in three different cases; one of which (#2) had dup(1q) with different breakpoints between two subclones. scWGS also revealed more complex patterns of copy number changes associated with structural events in #3 and #4. Further analysis with FISH and bulk WGS confirmed that these structural abnormalities involved complex rearrangements of chromosomes 16 and 14, respectively (Fig. 2, Supplementary Fig. 4). Thus, scWGS can also be used to delineate complex structural events.

Taken together, phylogenetic analysis of the scWGS data showed that most of the chromosomal gains were present at the root of the phylogenetic trees, with clonal evolution involving gains or losses of 1–2 chromosomes in approximately half of the cases. Structural changes, on the other hand, generally occurred later during leukemogenesis.

### Aneuploid pattern based on copy number changes in 577 cases reveals selective pressures

To elucidate further the aneuploid pattern in HeH ALL, we next studied copy number data derived from single nucleotide polymorphism (SNP) arrays, whole exome sequencing (WES), or WGS for 577 primary

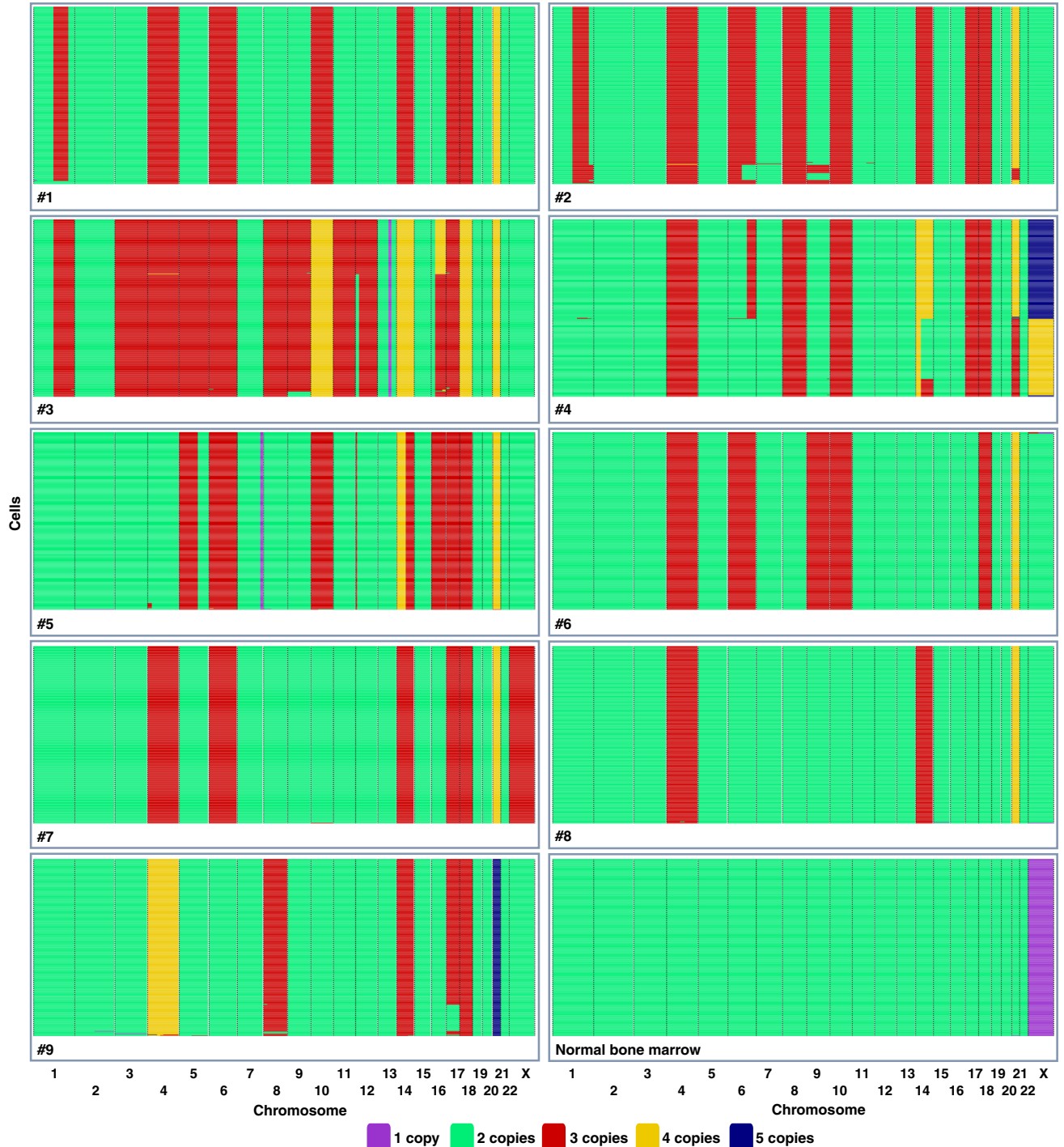

**Fig. 1 | Single cell whole genome sequencing results from nine primary high hyperdiploid childhood acute lymphoblastic leukaemia cases and one normal bone marrow.** The heatmaps show the genome-wide copy number of each individual cell with a resolution of 5 Mb (the Y chromosome is not included). Overall, only low to very low levels of copy number heterogeneity was seen. Created with BioRender.com. Source data are provided as a Source Data file.

cases (Supplementary Data 1, Supplementary Fig. 5), in total encompassing 13271 chromosomal pairs. Of these, 6 (0.045%) were monosomic, 8410 (63%) disomic, 3997 (30%) trisomic, 829 (6.2%) tetrasomic, and 29 (0.22%) pentasomic. Together, these data corroborate the view that HeH ALL is primarily characterized by trisomies and tetrasomies[5], with monosomies being exceedingly rare.

To be able to model HeH development, we utilized this copy number data to better understand selective pressures, reasoning that chromosomal gains providing a selective advantage are more common. Eight chromosomes were gained in more than 70% of cases:

chromosomes 21 (100%), X (97%), 14 (95%), 6 (89%), 18 (83%), 4 (82%), 17 (78%), and 10 (74%), indicating a strong selection for extra copies of these chromosomes and suggesting that these gains are highly likely driver events. Six additional gains were relatively common: chromosomes 8 (38%), 5 (23%), 9 (19%), 11 (14%), 12 (14%), and 22 (11%). These copy number alterations might also be (co-)driving events, at least occasionally. The remaining autosomal chromosomes were gained in <10% of the cases and hence unlikely to provide a selective advantage; some, such as chromosomes 13 and 20, which were recurrently monosomic, may even be selected against. Chromosome Y displayed

**Table 1 | Genetic heterogeneity in nine high hyperdiploid childhood acute lymphoblastic leukaemia cases based on single cell whole genome sequencing**

| Case | Number of cells sequenced | Number of clones (% of cells)[a] | Number of clones—numerical changes only (% of cells)[a] | Number of cells with a unique genome | Genome-wide heterogeneity score | % of cells with PCG |
|---|---|---|---|---|---|---|
| 1 | 269 | 2 (A, 98%; C, 1.9%) | 1 (A, C, 100%) | 1 | 0.07 | 22 |
| 2 | 257 | 6 (A, 88%; E, 3.9%; D, 2.7%; I, 1.9%; F, 1.2%; H, 0.8%) | 4 (A, 88%; E, 3.9%; F, H, I, 3.9%; D, 2.7%) | 5 | 1.20 | 85 |
| 3 | 272 | 4 (E, 64%; A, 30%; H, 2.6%; G, 0.7%) | 3 (E, G, 65%; A, 30%; H, 2.6%) | 7 | 1.16 | 14 |
| 4 | 348 | 5 (H, 55%; C, 34%; A, 8.9%; E, 0.9%; I, 0.6%) | 5 (H, 55%; C, 34%; A, 8.9%; E, 0.9%; I, 0.6%) | 5 | 5.11 | 19 |
| 5 | 347 | 2 (A, 96%; D, 2.9%) | 1 (A, D, 99%) | 3 | 0.18 | 20 |
| 6 | 271 | 2 (A, 99%; B, 0.7%) | 1 (A, B, 100%) | 0 | 0.04 | 5 |
| 7 | 273 | 1 (A, 100%) | 1 (A, 100%) | 1 | 0.12 | 10 |
| 8 | 266 | 1 (A, 99%) | 1 (A, 99%) | 3 | 0.16 | 0 |
| 9 | 269 | 4 (B, 75%; F, 15%; A, 7.1%; I, 0.7%) | 4 (B, 75%; F, 15%; A, 7.1%; I, 0.7%) | 7 | 0.69 | 10 |

*PCG* primary constriction gap.

[a]Letters correspond to different clones as denoted in Fig. 2.

both gains (21% of male cases)−always as XXYY or XXXYY−and nullisomy (4% of male cases), indicating that it is neutral to selection.

Recurrent tetrasomies were seen for chromosomes 21 (81%), X/Y (20%; including XXXX in females and XXXY/XXYY in males), 14 (17%), 18 (12%), 10 (8.3%), 8 (2.6%), and 4 (1.7%). The majority (787/829; 95%) of tetrasomies were of the 2:2 type, i.e. showed duplication of both chromosomal homologues. Of the 42 3:1 tetrasomies (triplication of one homologue and retention of the other), 31 (74%) were for chromosome 21 and five (12%) were XXXY. Apart from one case with XXXYY, pentasomy was only seen for chromosome 21 (4.5% of cases), indicating that the selection for extra copies of this chromosome is particularly strong.

UPIDs were seen in 208/577 (36%) of the cases (median 1/case, range 1–6). The UPIDs/all disomies ratio was 0–5% for all chromosomes except for chromosome 9, where it was 17%. This rather constant frequency (except for chromosome 9) suggests that, in general, UPIDs are passenger events.

**Subclonality indicates selective pressures**

Copy number analysis based on bulk samples has a limited resolution in detecting subclones, with an approximate detection limit of subclones corresponding to 20–30% of the cells (Supplementary Fig. 6). Nevertheless, subclonality involving relatively large clones that are detectable with this method can indicate ongoing clonal evolution and may reveal selective pressures in the leukaemic population. The majority (72%) of the 577 HeH ALLs did not have detectable subclonality involving whole chromosomes, agreeing well with the scWGS data. Most chromosomes displayed subclonality in <3% of cases, but higher levels were seen for chromosomes 8 (4.5%), 9 (8.7%), 21 (3.6%), and X in females (5.1%) (Supplementary Table 2). For chromosomes 8, 9, and X, subclonality was mainly seen between two and three copies, either in the form of (hetero)disomy/trisomy or in the form of UPID/trisomy; two forms of subclonality that have approximately the same detection limits in the HeH scenario. Whereas the former of these could arise either by an initial disomy becoming a trisomy or vice versa, the latter can only arise from initial trisomy by loss of one

chromosomal homologue (Supplementary Fig. 7). Then, the likelihood is 2/3 that it becomes a heterodisomy (normal disomy with retained heterozygosity) and 1/3 that it becomes a UPID. Most chromosomes conformed to the expected ratio of subclonal disomy/trisomy to UPID/trisomy (Supplementary Table 2), suggesting loss from trisomy. For chromosome X in females, however, trisomy/UPID subclonality was significantly more common than expected ($P = 2.60 \times 10^{-4}$; two-sided exact binomial test), which is likely explained by preferential loss of the inactive X, as it is usually the active X that is duplicated in HeH ALL with trisomy X[12]. Chromosome 8 displayed borderline significance ($P = 0.0529$; two-sided exact binomial test) for fewer cases with subclonal UPID/trisomy than expected (Supplementary Table 2), possibly indicating that some cases were gaining an extra chromosome from a disomy, in line with positive selection. Chromosome 9 displayed frequencies of subclonal disomy/trisomy and UPID/trisomy agreeing with loss from a trisomic state, indicating selection against trisomy. Finally, subclonality for chromosome 21 was mainly seen for trisomy/tetrasomy and tetrasomy/pentasomy, indicating selection for extra chromosomal copies. Altogether, selection against extra copies of chromosome 9 and for extra copies of chromosome 21 and possibly chromosome 8 was apparent, with the reservation that subclones corresponding to less than 20-30% of the cells could not be analyzed.

**Comparison of diagnostic and relapse samples shows positive selection for trisomy 8 and negative for trisomy 9**

Selective pressures can also be inferred from comparing paired samples obtained at different time points. We studied chromosomal copy number and ascertained whether trisomies and UPIDs involved the same chromosomal homologue in paired diagnostic/relapse samples from 23 cases. Such samples have previously been shown to be clonally related and display overall very similar karyotypes[15,16]. In total, 4.4% of 529 chromosomal pairs differed in copy number between the diagnostic and relapse samples (Supplementary Data 2). Of the 171 investigated trisomies and UPIDs, only one trisomy 8 involved different chromosomal homologues in the diagnostic and relapse sample, indicating that heterogeneity of this type is rare in HeH ALL.

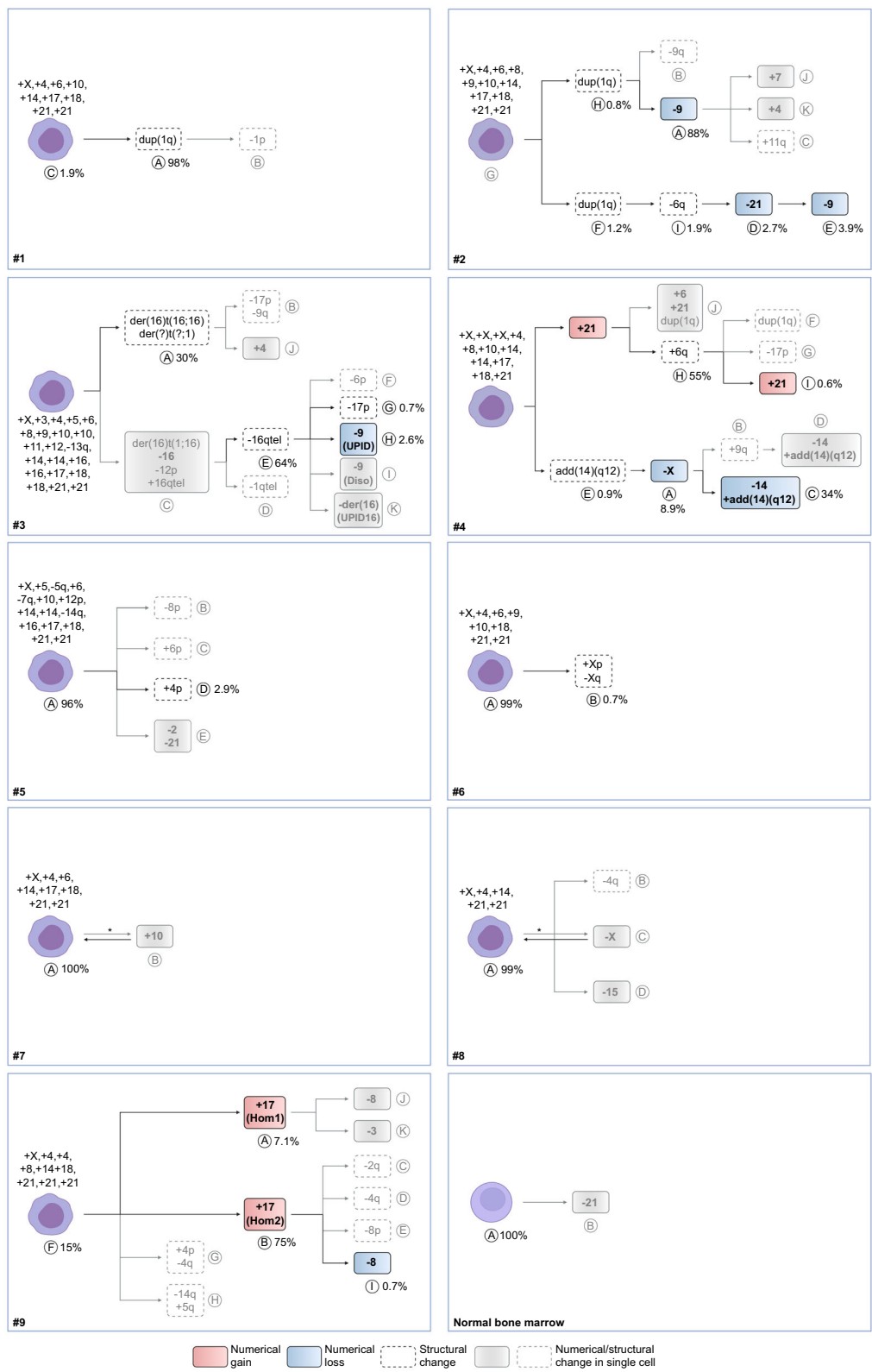

**Fig. 2 | Phylogenetic trees showing the most probable course of genetic evolution, based on single cell whole genome sequencing (scWGS), bulk WGS, and fluorescence in situ hybridization in nine primary childhood acute lymphoblastic leukaemia cases and one normal bone marrow.** The bulk of the chromosomal gains was present already in the inferred earliest cells, with 1–2 chromosomes being gained or lost during clonal evolution in some of the cases. *indicates that the direction of the clonal evolution cannot be determined. Diso heterodisomy, Hom1 homologue 1, Hom2 homologue 2, UPID uniparental isodisomy. Created with BioRender.com.

Chromosomes that recurrently differed between diagnostic and relapse samples were chromosomes 8 (17%), 4, 9, 21 (13%), and X, 7, 10, and 15 (8.7%). Trisomy 8 displayed signs of positive selection, as it never went from trisomy to UPID and as the trisomy involved different homologues in one case. Chromosome 9, on the other hand, displayed UPID in one sample and heterodisomy in the other in three cases, indicating an original clone with trisomy 9 that was selected against.

Altogether, analyses of the frequencies of chromosomal gains, subclonality patterns, and paired diagnostic/relapse samples suggested that the chromosomal gains in HeH ALL can be divided into three groups based on the selective pressures: chromosomes X, 4, 6, 10, 14, 17, 18, and 21, which are associated with strong positive selection (group strong-pos), chromosomes 5, 8, 11, 12, and 22, which are associated with weaker positive selection (group weak-pos), and chromosomes Y, 1–3, 7, 9, 13, 15, 16, 19, and 20, which are neutral or associated with negative selection (group neg).

### Simulation of HeH development suggests formation by a tripolar mitosis

To understand further how the aneuploidy in HeH ALL arises, we next simulated hyperdiploidy development in silico under different scenarios. We included five possible routes to aneuploidy that have been reported to occur in cancer[17]: (1) sequential gains in a diploid cell (diploid/sequential), (2) initial tetraploidy followed by chromosomal losses (tetraploid/sequential), (3) tripolar division in a diploid cell (diploid/tripolar), (4) tripolar division in a tetraploid cell (tetraploid/tripolar), and (5) mitotic catastrophe resulting from complete loss of sister chromatid cohesion (mitotic catastrophe) (Supplementary Fig. 8). For models 3, 4, and 5, the simulation started with an abnormal mitosis directly resulting in aneuploid daughter cells according to the respective mechanism, followed by a low likelihood of nondisjunction of individual chromosomes, whereas mechanisms 1 and 2 started with a diploid or tetraploid cell, respectively, followed by individual nondisjunction events. First, we only included positive selection for the strong-pos group of chromosomes, i.e. X, 4, 6, 10, 14, 17, 18, and 21, with gains of chromosome 21 given the highest selective advantage based on its ubiquitous presence in these leukaemias. Briefly, 50,000 virtual cells were followed over multiple generations, with gain of strong-pos chromosomes increasing survival probability in the daughter cells and other nondisjunction events lowering it. Since the UPID frequency in the patient cohort was constant at 2.5% for non-strong-pos chromosomes (except chromosome 9), simulations were stopped when this level was reached. The resulting virtual cell populations were then compared with the chromosomal patterns in the 577 primary HeH ALLs.

All models resulted in a continuous increase in the UPID frequency over generations (Supplementary Fig. 9A). For the diploid/tripolar and diploid/sequential models, UPID frequencies of 2.5% were reached after 50–800 generations (median 72.5 and 485, respectively) and for the tetraploid/sequential model within 10 generations. For the tetraploid/tripolar and mitotic catastrophe models, the initial UPID frequency was >2.5% (18.2% and 9.9%, respectively) and plateaued at >30% after 1000 generations. Since this was inconsistent with the patient data, they were removed from further testing.

Next, we investigated the average number of trisomies/tetrasomies at different modal chromosome numbers (MCN). Interestingly, a marked elevation change was observed at MCN 62 for trisomies in the patient cohort (Fig. 3a), indicating that there may be two subgroups with different trisomy:tetrasomy ratios: MCN 51–61 ($n = 545$) and MCN 62–67 ($n = 32$), respectively. This suggests that HeH ALL with lower and higher MCN could arise through different mechanisms. Therefore, we investigated these groups separately in the following analyses.

Starting with MCN 51–61, we observed that the tetraploid/sequential model resulted in very few such cells (Supplementary

Fig. 9B). We therefore concluded that this model could not give rise to HeH with MCN 51–61 and excluded it from further testing. We then compared the pattern of trisomies and tetrasomies at different MCN (Fig. 3a) and the pattern of trisomies and tetrasomies for each chromosome (Fig. 3b) between the HeH ALL patient data and the simulations results by the root mean squared error (RMSE) method (Supplementary Table 3). The diploid/tripolar and diploid/sequential models both showed low RMSE values, indicating that they fit relatively well with the patient data. We next looked at the frequency of tetrasomy 21 of the 2:2 type (duplication of both homologues) and 3:1 type (triplication of one homologue). In the diploid/sequential model, the 3:1 type was enriched during the simulation process, resulting in 64% tetrasomy 21 of this type. However, the patient data and the diploid/tripolar model both showed lower proportions of tetrasomy 3:1 (6.6% and 21%, respectively), supporting a diploid/tripolar origin. Notably, 3:1 tetrasomies were not an indication of one homologue being selected for, but rather resulted from the strong overall selection for extra copies of chromosome 21 in both models. To see if we could fine-tune the diploid/tripolar further, we included positive selection also for the weak-pos chromosomes. The modified version yielded even lower RMSE values than the original one (Supplementary Table 3). Hence, our simulations showed that the diploid/tripolar model consistently resulted in virtual cells with karyotypes similar to those seen in HeH ALL with MCN 51–61. Furthermore, sampling of the simulation results over consecutive generations showed that the diploid/tripolar model displayed whole chromosome copy number evolution consistent with a punctuated evolution model, with an initial sharp rise in chromosome numbers (Supplementary Fig. 10).

Next, we turned to the MCN 62–67 group, again studying the pattern of trisomies and tetrasomies at different MCN and for different chromosomes. Here, both the diploid/tripolar and the tetraploid/sequential models agreed well with the patient data (Supplementary Table 3), with the tetraploid/sequential model more closely following the distribution of average number of trisomies and tetrasomies across MCNs (Fig. 3a). We included selection for the weak-pos chromosomes also here and, since the UPID frequency is higher at higher MCNs, let the simulations run to a UPID frequency of 5%. Both the diploid/tripolar and tetraploid/sequential models resulted in virtual cells that fit well with the patient data (Supplementary Table 3). Interestingly, looking at the patient copy number data, cases with MCN 62–67 had more subclonality, with half of these cases (16/32) harbouring ≥1 subclonal chromosome; significantly higher than observed in the other HeH ALLs ($P = 0.0006$; two-sided Fisher's exact test). Furthermore, the only case in the scWGS analysis with MCN in this range (#3) also had a relatively high heterogeneity score.

Taken together, our modelling in conjunction with the patient data suggested that, in most instances, high hyperdiploidy in paediatric ALL arises by a tripolar division in a diploid cell. However, HeH ALL with MCN 62–67 (comprising around 5% of cases) may possibly arise by initial tetraploidy followed by chromosomal losses.

### Chromosomal age pattern validates a tripolar division origin

In the diploid/tripolar model, most chromosomes are gained in the initial mitosis but some are gained and fixed during clonal selection. Seeking to validate our results from the in silico modelling, we reasoned that chromosomes that are gained later during clonal evolution should primarily be those that give a selective advantage, i.e. the strong-pos and weak-pos groups. In contrast, neg chromosomes would all have been gained at the initial division since they would not be selected for, although some may arise later due to drift. Therefore, we hypothesized that strong-pos and weak-pos trisomies should, on average, be newer than neg trisomies. If the hyperdiploidy instead arose by sequential gains, there would be no difference in the ages of the trisomies between these groups, as they could arise in any order (Fig. 4a).

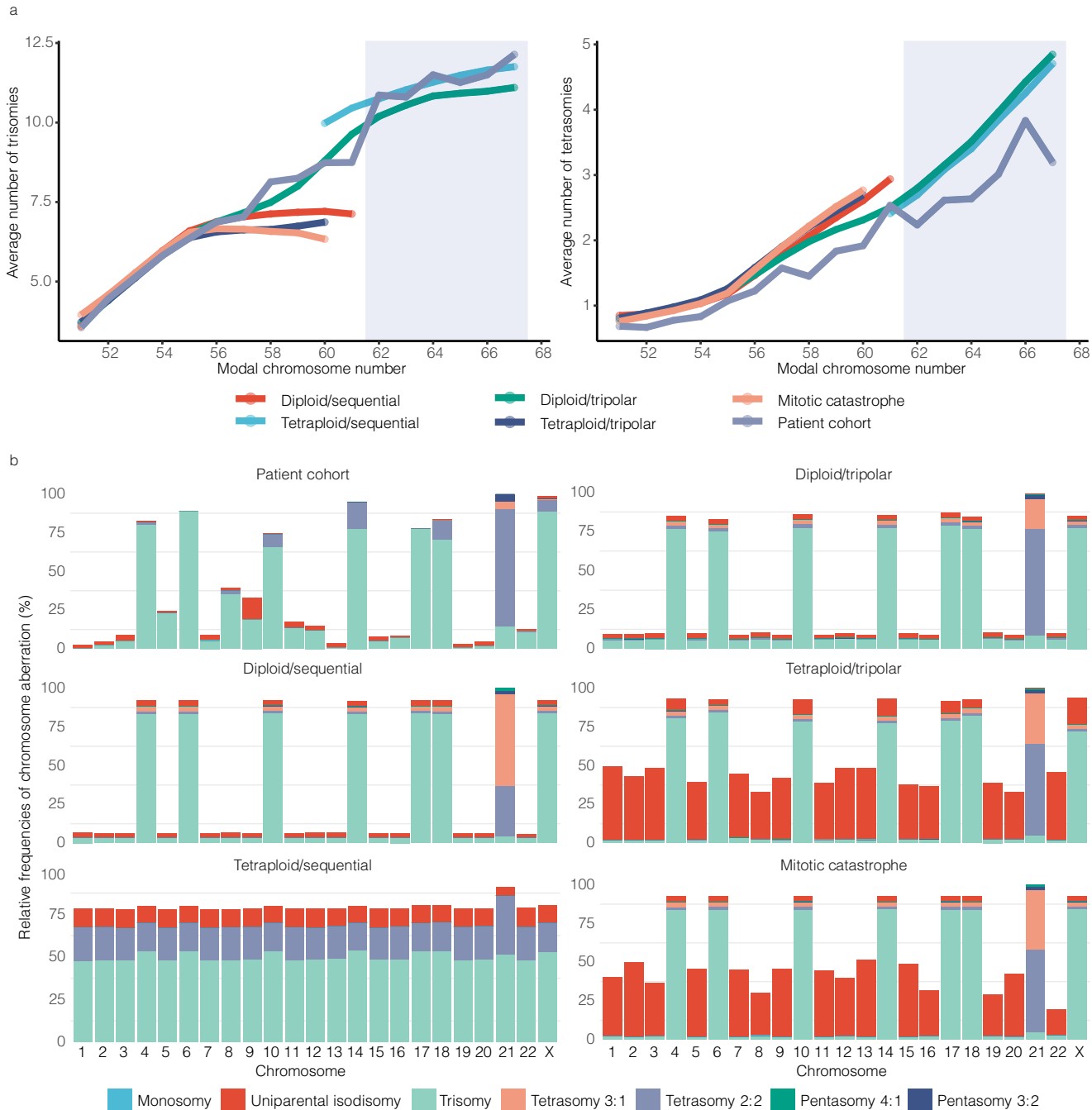

**Fig. 3 | Simulation of high hyperdiploidy development in childhood acute lymphoblastic leukaemia according to five different models: (1) sequential gains in diploid cell (diploid/sequential), (2) initial tetraploidy followed by chromosomal losses (tetraploid/sequential), (3) tripolar division in a diploid cell (diploid/tripolar), (4) tripolar division in a tetraploid cell (tetraploid/tripolar), and (5) mitotic catastrophe resulting from complete loss of sister chromatid cohesion (mitotic catastrophe).** Data shown are from the end point in the simulations. **a** Correlation between the average number of trisomies/tetrasomies and the modal chromosome number (MCN) in the simulation results and the patient cohort of 577 cases of high hyperdiploid ALL. The average number of trisomies at each modal number in the diploid/tripolar model closely follows what is seen in the patient cohort at MCN 51–61, indicating a very good fit of the model to patient data. At MCN 62–67, there is a sharp increase in the average number of trisomies per modal number in the patient cohort (indicated by a grey square), and it follows the tetraploid/sequential model more closely, possibly indicating a

different mechanism. The average number of tetrasomies at each modal number in the patient cohort is based on fewer chromosomes (since tetrasomies are less common than trisomies) and follows most closely the diploid/tripolar and the tetraploid/sequential models for MCN 51–61 and MCN 62–27, respectively. **b** Pattern of chromosomal copy number changes and uniparental isodisomies resulting from the simulations according to each model and in the patient cohort. Frequency of each type of aberration (as given in the legend) is seen on the Y axis and each chromosome (except Y) on the X axis. Whereas the tetraploid/sequential, tetraploid/tripolar, and mitotic catastrophe model all result in chromosomal patterns very different from the one seen in the patient cohort, the diploid/sequential model, diploid/tripolar model, and patient cohort display relatively similar patterns. However, based on the high frequency of 3:1 tetrasomies in the diploid/sequential model it could be excluded, leaving the chromosomal pattern resulting from the diploid/tripolar model most similar to the one seen in the primary cases. Source data are provided as a Source Data file.

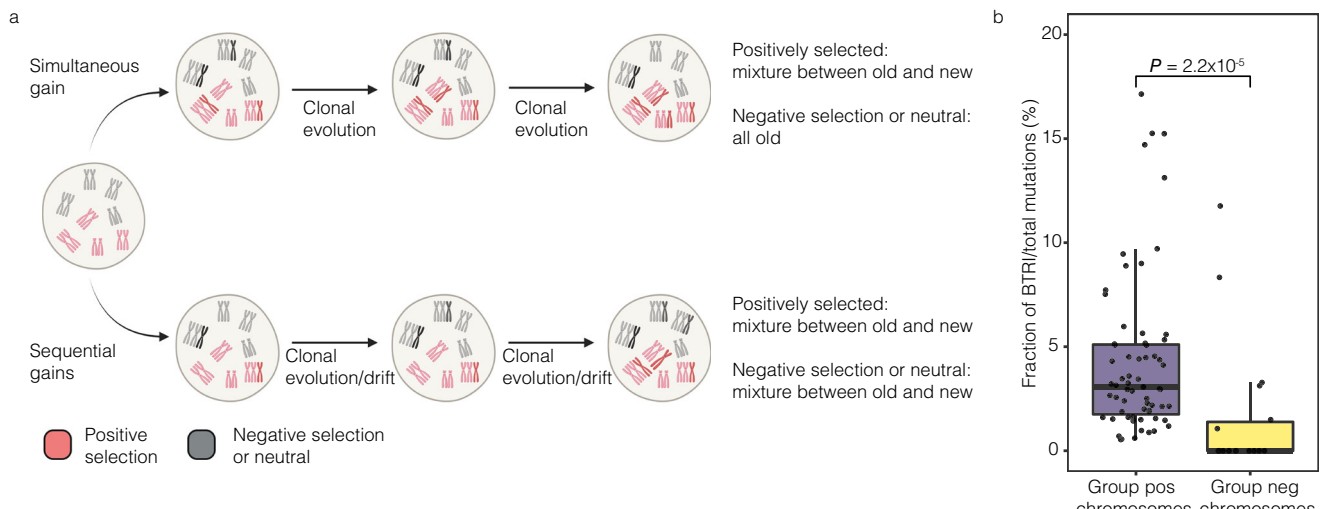

**Fig. 4 | Analysis of somatic single nucleotide variants (SNVs) in trisomic chromosomes supports a simultaneous gain of most chromosomes followed by clonal evolution in high hyperdiploid (HeH) childhood acute lymphoblastic leukaemia (ALL). a** Schematics of chromosomal gains in a scenario where chromosomes are either gained predominately by a first hit or via sequential nondisjunction. Trisomies subjected to positive selection (strong-pos and weak-pos trisomies) are shown in red and trisomies subjected to negative selection or neutral (neg trisomies) in grey. These two scenarios are expected to result in different mixtures of older and newer chromosomes. **b** Boxplots of the fraction of BTRI mutations in groups strong-pos/weak-pos versus group neg trisomies in 67 cases of HeH ALL. Groups strong pos/weak trisomies have a higher fraction of BTRI mutations ($P = 2.2 \times 10^{-5}$ Mann−Whitney two-sided test), indicating that they are on average newer, consistent with an initial tripolar cell division followed by clonal evolution. The centre of the boxplot is the median and lower/upper hinges correspond to the first/third quartiles; whiskers are 1.5 times the interquartile range and data beyond this range are plotted as individual points. Figure 4a was created with BioRender.com. Source data are provided as a Source Data file.

To investigate the age of different trisomies, we studied somatic single nucleotide variants (SNVs) in trisomies based on WGS in 67 HeH ALL. We utilized that SNVs that are present already before the trisomy forms (BTRI mutations) will be duplicated if they are in the gained homologue and display variant allele frequencies (VAFs) of -0.67. Conversely, SNVs that arise after the trisomy or in the homologue that is not duplicated will only be present in one of the homologues and display VAFs of -0.33 (B/ATRI mutations) (Fig. 5a, b). Hence, the proportion of SNVs that are of the BTRI type will be higher the newer the trisomy is, since the chromosome will have spent a longer time as not duplicated, allowing time for more mutations to arise. Among the 67 investigated cases, 536 of 15,828 SNVs (3.39%) in groups strong-pos and weak-pos chromosomes were of the BTRI type and 9 of 819 (1.09%) in group neg chromosomes ($P = 2.2 \times 10^{-5}$; Mann−Whitney two-sided test) (Fig. 4b). Thus, the former chromosomal gains were on average newer than the remaining trisomies, in line with what would be expected from a diploid/tripolar origin.

**Mutational signatures show different etiological factors during leukemogenesis**

To gain further insight into the leukemogenesis of HeH ALL, we studied mutational signatures in 67 cases with bulk WGS data. We investigated BTRI and B/ATRI mutations in trisomic chromosomes and relapse-specific mutations, since these groups can be put into a distinct timeline (Fig. 5a, b). BTRI mutations were predominantly associated with mutational signatures SBS1 and SBS5 (Fig. 5c); known clock-like signatures likely caused by intrinsic mutational processes[18,19]. Their high frequency at the earliest time point, before the hyperdiploidy arises, agrees well with an early origin devoid of environmental exposure. B/ATRI mutations displayed a wider range of mutational signatures, with SBS1, SBS5, SBS7a, SBS8, SBS18, SBS19, and SBS39 all contributing (Fig. 5c). Of these, SBS7a has been associated with ultraviolet light exposure[20]; this signature has previously been reported to dominate in some cases of aneuploid childhood ALLs[10,21] and it was present in six (9.0%) cases. SBS8 has been suggested to be associated with late replication errors[22], whereas SBS18 has been linked to

mutagenesis by reactive oxygen species[23]. SBS19 and SBS39 have unknown etiologies[20]. Mutations specific for the relapse samples, which represent the latest mutations, were similar to the B/ATRI mutations, but with addition of signatures SBS15, SBS26, and SBS87 (Fig. 5c), as has previously been reported for the TARGET cohort[24]. SBS15 and SBS26 are associated with defective DNA mismatch repair[24], whereas SBS87 is associated with thiopurine treatment and hence likely induced by chemotherapy[24].

**Temporal analysis of additional somatic events shows that the chromosomal gains are early**

To determine when other somatic genetic events occur in relation to the chromosomal gains, we analyzed structural rearrangements, deletions, and mutations, focusing on (1) subclonality and (2) events occurring in gained chromosomes or UPIDs, where the temporal order could be investigated by looking at the allelic patterns.

For structural rearrangements, the analysis comprised known drivers that can be identified from copy number data: dup(1q), deletions of 6q [del(6q)], isochromosomes 7q [i(7q)], and partial gains of 17q (gain_17q)[25,26]. Of these, dup(1q), del(6q), and gain_17q were frequently subclonal (30−40% of cases), whereas i(7q) was generally present in the main clone (Supplementary Table 4). One case had two different subclonal dup(1q), similar to #2 in the scWGS analysis (Fig. 2). Furthermore, analysis of BTRI and B/ATRI mutations showed a significantly higher proportion of BTRI mutations in dup(1q) than in trisomies (median 29.4% vs. 4.3%; $P = 4.9 \times 10^{-4}$; Mann−Whitney two-sided test), indicating a later origin (Supplementary Fig. 11). Temporal order could be determined for dup(1q) and del(6q), showing that 8/8 and 22/22 informative cases, respectively, arose after the UPID or chromosomal gain (Supplementary Table 4).

Deletions of *IKZF1*, *CDKN2A*, *PAX5*, *ETV6*, *CREBBP*, and *TCF3*[26,27] were subclonal in 10−40% of the cases (Supplementary Table 4). Temporal analysis showed that 15/16 *CDKN2A* deletions, 1/1 *PAX5* deletion, 8/9 *ETV6* deletions, and 1/1 *CREBBP* deletion occurred after the respective UPID or trisomy. Thus, most informative deletions happened after the respective chromosome became

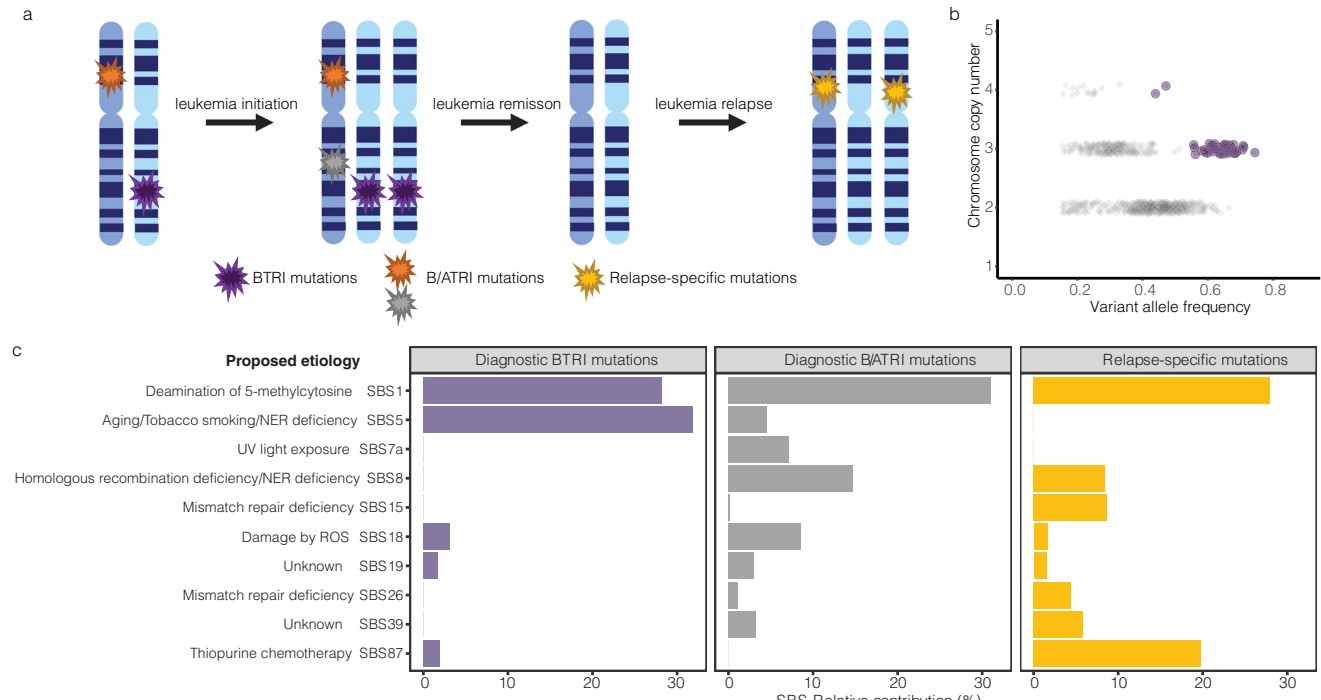

**Fig. 5 | Patterns of somatic mutations in high hyperdiploid acute lymphoblastic leukaemia. a** Timeline of mutations arising either before the trisomies (BTRI mutations), before/after the trisomies (B/ATRI mutations), and at relapse. **b** Variant allele frequencies (VAFs) in bulk sequencing data of scWGS case 1 for all mutations in relation to copy number based on the number of reads for that particular chromosome segment. BTRI mutations, which are present in 2/3 chromosomal homologues and have VAFs of -0.67, are shown in purple and B/ATRI mutations,

which are present in 1/3 chromosomal homologues and have VAFs of -0.33, are shown in grey. **c** Relative contribution of the ten most common SBS signatures in BTRI mutations in trisomic chromosomes, B/ATRI mutations in trisomic chromosomes, and mutations specific for relapse samples. NER nucleotide excision repair, ROS reactive oxygen species, SBS single base substitutions, UV ultraviolet. Figure 5a was created with BioRender.com. Source data are provided as a Source Data file.

trisomic, but one somatic *CDKN2A* deletion and one *ETV6* deletion (which we have previously shown to be constitutional[26]) occurred at a disomic state.

Finally, we looked at 338 driver mutations in 218 cases where WES or WGS data were available. Of these, 150 (44%) were subclonal (Supplementary Tables 4 and Supplementary Data 3). For clonal mutations in trisomies, tetrasomies, or UPIDs, mutations were also classified as B/ATRI or BTRI. Sixty (92%) B/ATRI mutations and five (8.7%) BTRI mutations were found, including one *IKZF1* mutation in a case with UPID7 (Supplementary Data 3).

Taken together, the analysis showed that structural rearrangements, deletions, and mutations were frequently subclonal and generally occurred after the chromosomal event, supporting an overall scenario where the hyperdiploidy arises first and other somatic aberrations occur at later stages.

## Discussion

We have performed a detailed analysis of the genetic mechanisms and the temporal order of different genetic events in HeH ALL. Using a combination of scWGS and in-depth analysis of SNP array and sequencing data from a large cohort of cases, we show that the chromosomal gains are early events and relatively stable throughout leukaemia development, whereas structural rearrangements and mutations generally occur later. In silico simulations of high hyperdiploidy development suggested that an initial tripolar division in a diploid cell, followed by clonal selection, best recapitulated the chromosomal patterns seen in patient samples.

Whether HeH ALL exhibits CIN has been debated. Cytogenetic data as well as bulk copy number analysis with SNP arrays have suggested that these leukaemias generally are chromosomally stable, with most cells displaying the same chromosomal gains[5,26,28]. However,

cytogenetic analyses only comprise the dividing cells and misclassification of chromosomes can lead to underestimation of heterogeneity, whereas SNP arrays cannot detect all minor clones. Several previous studies have also used interphase FISH to investigate chromosomal heterogeneity[14,29–31], but with variable results and conclusions, likely due to a high degree of technical artifacts[5]. Here, we used scWGS to circumvent the above problems. This method is superior for characterizing copy number heterogeneity by including all cells–also non-dividing–and due to unequivocal identification of chromosomes[32]. We found relatively little chromosomal heterogeneity, with non-clonal numerical changes seen in only 12 (0.47%) of all 2572 leukaemic cells sequenced and 5/9 cases having identical chromosomal content in >99% of the cells (Fig. 1). Of the remaining four cases, three displayed subclones that were also detectable by SNP array analysis, and only one appeared to have a single clone by SNP array analysis when in fact it had several minor clones. Thus, scWGS strongly supports that HeH ALL is chromosomally stable, in line with cytogenetic and SNP array data. Notably, this also shows that aneuploidy in cancer does not lead to CIN per se; something that has also been debated[33,34].

Although HeH ALL overall appeared stable, there was nevertheless some variation in heterogeneity between cases. We recently reported a high but varying frequency of sister chromatid cohesion defects in HeH ALL, possibly associated with low levels of cohesin and/or condensin[14]. Case 2 had cohesion defects in 85% of the metaphase cells, possibly explaining the high heterogeneity in this case. However, the remaining eight cases all had percentages of cells displaying cohesion defects that were at or below the median value (21%) in our previous study[14] (Table 1). Thus, it is possible that we would have found more chromosomal heterogeneity by scWGS if more cases with severe cohesion defects had been included in this study.

Several mechanisms have been suggested for how the extra chromosomes in HeH ALL are gained, including one abnormal mitosis, loss of chromosomes from a tetraploid cell, sequential gains due to CIN, and, most recently, fusion of a mitotic cell and a G0/G1 cell[11–13,29,35]. Any such mechanism should conform to/explain a number of features of HeH ALL genomes: (1) the specific pattern of trisomies, tetrasomies, and low-level UPIDs, including why 2:2 tetrasomies are much more common than 3:1 tetrasomies, (2) the presence not only of the common trisomies but also of gains (at low frequency) of all chromosomes, (3) the relative chromosomal stability shown by our scWGS analysis, and (4) that strong-pos and weak-pos chromosomes are on average newer (occur later during leukemogenesis) than neg chromosomes, as evidenced by our analysis of B/ATRI and ATRI mutations. To test which of the proposed mechanism(s) that conformed to the first of these features, we performed in silico modelling (excluding the fusion model since its outcome could not be statistically predicted) and compared the outcome with the chromosomal patterns seen in a large cohort of HeH ALL. We found that an initial tripolar mitosis that leads to gain of the bulk of the extra chromosomes, followed by clonal evolution over multiple generations of cells, recapitulated the chromosomal and allelic patterns seen in the patient samples. Furthermore, this mechanism can also explain why the low frequency trisomies occur, as they are passenger events that are gained in the initial tripolar division, as well as why they are on average older than the high frequency trisomies, which may also arise and be fixated later due to positive selection pressure. Finally, the diploid/tripolar model does not require chromosomal instability for aneuploidy to occur within a reasonable (considering the young age of the patients) time frame, as the bulk of the chromosomal gains occur very early (also in line with previous data showing hyperdiploidy years before overt diagnosis of HeH ALL[6–10]). Notably, tripolar cell divisions have been reported to occur in cancer and lead to viable daughter cells[4,36] that potentially could regain mitotic stability by clustering or loss of supernumerary centrosomes. Thus, no evidence of this initial mitotic error apart from the allelic patterns would still be visible at the time of diagnosis.

The punctuated evolution model in cancer states that somatic aberrations arise in short bursts of time very early in tumour evolution[1]. By scWGS, all cases showed phylogeny in line with this, with few intermediate cells indicating gradual evolution, long truncal distances, and short branching distances (Supplementary Fig. 3). The diploid/tripolar model that we suggest underlies the extra chromosomes in HeH ALL is a clear example of a way that such punctuated evolution for whole chromosome copy number changes could occur. Our results thus support previous studies showing frequent punctuated evolution for copy number changes in malignancies[3,4].

We found a possible difference in the chromosome distribution in HeH cases with MCN 62–67. Heerema et al.[37] reported that cases with MCN 63-67 have different chromosomal gains than HeH ALL with lower MCN, in line with them being a separate entity genetically. Furthermore, we and others have previously shown that cases with higher MCN have a significantly better prognosis[38,39], indicating that they also differ clinically. However, it should be noted that due to the rarity of cases with MCN in this span, we cannot exclude that the observed differences in chromosome distribution were due to chance only. Both a diploid/tripolar and a tetraploid/sequential mechanism agreed relatively well with the chromosomal patterns in the patient cohort, and further studies are needed to ascertain how HeH ALL with MCN 62–67 arises.

In conclusion, we present a model for the leukemogenesis of HeH paediatric ALL where most cases are initiated by an erroneous tripolar mitosis, after which they undergo low-level clonal evolution to optimize their chromosomal pattern and gain additional driver events that eventually leads to overt leukaemia several years later. This model agrees well with a wealth of previous observations, including the early

occurrence of the chromosomal gains[6–10], chromosomal and allelic patterns[11–13], and general genomic stability[5,26,28] in this disease. Furthermore, it strengthens the evidence that copy number changes and aneuploidy frequently arise by punctuated evolution at the early stages of tumorigenesis and that aneuploidy-driven malignancies do not necessarily have high levels of chromosomal copy number heterogeneity and CIN.

## Methods

### Single cell WGS

All investigations complied with relevant ethical regulations. Written informed consent was obtained from the patients and/or their guardians according to the Declaration of Helsinki and the study was approved by the Ethics Committee of Lund University, Sweden. No monetary compensation was offered for patient participation. Viable bone marrow cells obtained at diagnosis from nine patients with high hyperdiploid ALL and one healthy individual, selected on the basis of sample availability, were subjected to low-pass scWGS. Single nuclei in $G_0/G_1$ phase were isolated using a fluorescence-activated cell sorting (FACS) cytometer and DNA libraries were constructed for multiplexed whole genome sequencing with average sequencing depth between 0.006x to 0.089x per cell (median 0.02x)[40]. Sequencing reads were aligned to the UCSC human reference genome (hg19, [http://hgdownload.cse.ucsc.edu/goldenPath/hg19/bigZips/]) using the Burrows-Wheeler Aligner (BWA, v0.7.17)[41]. The aligned reads were sorted and merged with SAMtools (v1.9)[42]. The copy number state of each chromosome was determined using AneuFinder (v1.14)[32]. Briefly, duplicate reads, low-quality alignments (MAPQ < 20), and reads falling into the regions specified by the blacklists provided by AneuFinder were discarded. Read counts in 2.5 Mb, 5 Mb, and 10 Mb variable-width bins were GC-corrected and copy number states were determined using the edivisive algorithm with copy-number states nulli-, mono-, di-, tri-, tetra-, penta-, and hexasomy. The copy number state was also determined by Ginkgo[43] using default settings with a bin size of 1 Mb. All data were manually curated and in the final heatmaps, breakpoints were aggregated depending on supportive data from WGS, SNP array, and FISH. scWGS phylogenetic trees were constructed using MEDICC2[44] and subsequently manually curated to accommodate structural rearrangements by combining scWGS, WGS, SNP array, and FISH results for some cases. Pairwise distances of single cells and simulated normal diploid cells were calculated using Manhattan distance by R (version 4.1.2) to obtain a distance matrix for each tumour. Phylogenetic inference for single cell trees and consensus trees were performed with the balanced minimum evolution algorithm from R package ape (v5.6)[45]. Normal diploid nodes for phylogenetic tress were constructed from simulated variable binning profiles in which bins presented an integer copy number equal to 2 for autosomes and 1/2 for chromosome X depending on patient sex. Clones were defined as ≥2 cells presenting with the same numerical and/or structural aberrations. Genome-wide heterogeneity scores were obtained from AneuFinder. Homologue inheritance of chromosomes gained or lost was determined by screening for heterozygous variants identified from bulk WGS data. Briefly, heterozygous variants were called by GATK (v4.0.11.0) haplotypecaller[46] and the variants from trisomies and tetrasomies of 3:1 type and UPIDs were extracted. For trisomies/tetrasomies 3:1, heterozygous variants were assigned to different homologues based on the alternative allele frequency obtained from bulk WGS data. Variants with alternative allele frequency higher than 0.6 were assigned to one chromosomal homologue and variants with alternative allele frequency less than 0.4 were assigned to the other. For UPIDs that were found in diagnostic samples, remission-specific variants from the same chromosome were assigned to one chromosomal homologue and variants that showed heterozygosity in the remission sample but homozygosity in the matched diagnostic sample were assigned to the other homologue. Then variants informative for

chromosomal homologue were screened in scWGS data and the homologue inheritance of chromosomes gained or lost was determined by the ratio between the number of each type of variant in the single cells.

## Copy number analysis of bulk data

Log R ratio (LRR) and B allele frequency (BAF) of SNP array data from Illumina (.idat files) and Affymetrix (.CEL files) intensity files were analyzed by Illumina GenomeStudio (v2.0, Illumina, San Diego, CA) and Affymetrix Analysis Power Tools (v2.10.0, Thermo Fisher Scientific Inc., Waltham, MA), respectively. Copy number alterations were called using TAPS[47] and manually reviewed in GenomeStudio or Chromosome Analysis Suite (v3.3, Thermo Fisher Scientific Inc., Waltham, MA). Subclonality of whole chromosomes was assessed using the TAPS software from SNP array, WES, or WGS data, considering LRR, BAF, and tumour purity. Depending on the type of subclonality (disomy/trisomy, UPID/trisomy, etc.), the lower limit of detection of subclones was estimated to 20-30% of the cells. The dataset included four different cohorts: from our Department[26], Zaliova et al.[48], Duployez et al.[49], and The Therapeutically Applicable Research to Generate Effective Treatments (TARGET) program (dbGAP accession number phs00464) (Supplementary Data 1). Of those 577 cases, 253 (44%) were females and 324 (56%) were males, based on the absence or presence of a Y chromosome.

For WES data from TARGET, paired-end reads were aligned to the human reference genome hg19 by the bwa[41]. Duplicate reads marking and local realignment were performed by GATK[46]. Constitutional variants of matched tumour/normal pairs were called by GATK HaplotypeCaller and the bedtools (v2.27.1) intersect was used to extract the variants in the regions targeted by the exome sequencing kit. After normalizing read counts of constitutional mutation sites to the sequencing depth, the LRR of the constitutional variants was then calculated by the log-odds ratio of the variant allele count in the tumour versus in the normal. Reference allele frequency of constitutional variant sites was defined by the reference allele count versus total sequencing depth of the constitutional variant site in the tumour sample.

Paired diagnostic and relapse samples have been previously published[16] or were from TARGET. To investigate the chromosomal homologue involved in paired diagnostic and relapse samples, heterozygous variants from trisomies and tetrasomies 3:1 were extracted and assigned to different homologues based on the BAF of the diagnostic sample. Variants with BAF higher than 0.6 were assigned to one chromosomal homologue and variants with BAF less than 0.4 were assigned to the other. Then variants informative for chromosomal homologues were screened in the relapse sample to determine the involved chromosomal homologue. For UPIDs that were found in diagnostic samples, variants with BAF higher than 0.8 were screened in the paired relapse sample and homologue inheritance of chromosomes was determined by the BAF of corresponding variants in the relapse sample.

## WGS data analysis and identification of BTRI and B/ATRI mutations

WGS data from 14 BCP ALL cases have been previously published[10]. The initial putative somatic mutations were identified by the Complete Genomics Cancer Sequencing pipeline and the data were further filtered for Somatic Score ≥0 and number of unique reads for the mutated allele >10. For Complete Genomics data generated by the TARGET program (n = 34), somatic variants were identified by the TARGET WGS analysis pipeline ([https://ocg.cancer.gov/programs/target/target-methods#3233]). Illumina WGS sequencing libraries of nineteen matched diagnostic and remission bone marrow or peripheral blood samples diagnosed at Skåne University Hospital, Sweden, were constructed by the TruSeq Nano DNA sample preparation kit

(Illumina, San Diego, CA, USA). Paired-end sequencing (2x150bp) was done to ~60x coverage for diagnostic samples and ~30x coverage for remission. Somatic variants were identified by the GDC DNA-Seq analysis pipeline ([https://docs.gdc.cancer.gov/Data/Bioinformatics_Pipelines/DNA_Seq_Variant_Calling_Pipeline/]). Whether mutations occurred before (BTRI) or before/after (B/ATRI) trisomy formation were determined by mutant allele fractions according to Paulsson et al.[10]. Driver genes/mutations were identified by MutsigCV[50] and DriverPower[51]. A literature review focusing on genes identified by bulk WGS sequencing as targeted by non-silent somatic mutations associated with the search terms "aneuploidy", "instability" and "cohesin" was performed in order to investigate whether mutations in genes affecting genomic stability were responsible for the heterogeneity observed within the nine scWGS cases.

## Mutational signatures analysis

The R package MutationalPatterns[52] (v3.4.1) was used to decompose mutational profiles into pre-defined single base substitution (SBS) mutational signatures based on the Sanger mutational signatures (v3.2 - March 2021) and to ascertain the relative contributions of the SBS mutational signatures for BTRI and B/ATRI mutations in trisomic chromosomes at diagnosis, and all informative relapse-specific mutations.

## Cohesion assay and FISH

Sister chromatid cohesion was analyzed in metaphase spreads in all nine HeH ALL patient samples subjected to scWGS. The percentage of cells with cohesion defects, measured as visible primary constriction gaps (gaps between the sister chromatids at the centromeres)[14], was counted. FISH metaphase spreads mounted with DAPI were used for the assay, where 20–39 cells were analyzed per case. Images were captured using a Z2 fluorescence microscope (Zeiss, Germany) and the CytoVision software (v7.4, Leica, Germany).

Metaphase FISH was carried out on cases 2, 3, and 4 according to standard methods, with a total of 17–35 cells captured for each analysis. All whole chromosome paint FISH probes were acquired from Applied Spectral Imaging (Carlsbad, CA), and locus-specific probes from Vysis (Abbot Laboratories, Chicago, IL). FISH analysis was performed as follows: slides from case 2 were hybridized with whole chromosome paint probes for chromosomes 1 (Aqua – blue), 6 (Cy3 – red), and 21 (FITC – green); for case 3, whole chromosome paint probes for chromosomes 1 (FITC) and 16 (Aqua) were used together with a telomeric probe for 16q (Cy3); and for case 4, one analysis was performed with whole chromosome paint probes for chromosomes 3 (FITC) and 6 (Cy3), and another analysis for chromosome 14 (Aqua) together with a LSI TRA/D (14q11.2) break-apart dual colour probe (Cy3/FITC).

## Simulation of high hyperdiploidy development

To investigate the development of aneuploidy observed in HeH ALL, we constructed an algorithm to simulate the clonal expansion and to trace single-cell karyotypes over two thousand generations using the Python programming language (v2.7.15). For each model, 50,000 virtual cells were created and the copy number of individual chromosomes was defined according to the initial hit based on the simulation model: sequential gains in a diploid cell (diploid/sequential), initial tetraploidy followed by chromosomal losses (tetraploid/sequential), tripolar division in a tetraploid cell (tetraploid/tripolar), tripolar division in a diploid cell (diploid/tripolar), and mitotic catastrophe (mitotic catastrophe). For simplicity, all scenarios started with 46,XX cells (the Y chromosome was not included in the analysis). All virtual cells were represented by a $23 \times 50,000$ matrix. For the virtual cells ($C_g$) at generation $g$, $C_g(i)$ was the copy number of a virtual cell for each of the 23 chromosomes indexed by $i$. During cell division, two daughter cells would be formed from the mother cell. The missegregation rate ($M_{misseg}$) was set to ($15 \times 10^{-4}$/chromosome/

mitosis)[53] and the probability of missegregation ($P_{misseg}$) of each chromosome was weighted by the copy number of the given chromosome ($N$) and $P_{misseg} = M_{misseg}N$. Only one missegregation event of any given chromosome in a single cell division was allowed and the missegregated chromosome was randomly assigned to one of the two daughter cells. For tetraploid/sequential, the probability of chromosome loss was set to 35% according to previously published data[54]. Virtual cells with nullisomy were excluded from subsequent generations. Clonal expansion of virtual cells was altered by positive and negative selection of gain/loss of certain chromosomes. In the algorithm, we employed a survival/proliferation score ($S_{score}$) to determine the survival probability of virtual cells. Normal diploid cells were given a probability of 50% for proliferative survival. The $S_{score}$ of the virtual cell was determined according to its karyotype. Virtual cells with trisomies X, 4, 6, 10, 14, 17, and 18 (group 1) and gain of chromosome 21 (group 2) were subjected to positive selection, whereas virtual cells with gain/loss of the remaining chromosomes (group 3) were subjected to negative selection. In addition, virtual cells were also subjected to negative selection pressure (aneuploidy penalty score, $S_{aneuploidy}$), which increased with the modal number of chromosomes ($MCN > 46$) of that cell. The $S_{score}$ of the given virtual cell was computed according to:

$$S_{score} = 0.5 + \frac{NT_{g1} + 2NT_{g2} - NT_{g3}}{23} - S_{aneuploidy} \quad (1)$$

where $NT_{g1}$ is the number of trisomic chromosomes in group 1, $NT_{g2}$ is the number of trisomic chromosomes in group 2 and $NT_{g3}$ is the number of trisomic chromosomes in group 3. The $S_{aneuploidy}$ was calculated by using the probability density function of beta distribution from python scipy package (https://docs.scipy.org/doc/scipy/reference/generated/scipy.stats.beta.html) with the empirically determined location parameter loc 0, scale parameter scale 1.8, shape parameters a and b 0.18, 0.65, respectively. The $x$ parameter was defined as:

$$x = \frac{MCN - 46}{46} \quad (2)$$

For the tetraploid/sequential model, no $S_{aneuploidy}$ was used since no initial tetraploid cell would survive under that condition.

In addition, an extended version (four groups version) of $S_{score}$ was also used by dividing group 3 into two groups: one with negative selection for gain of chromosomes 1–3, 7, 9, 13, 15, 16, 19, and 20 (group 3b) and the other one with weak positive selection for gain of chromosomes 5, 8, 11, 12 and 22 (group 4). Then the $S_{score}$ of the given virtual cell was computed according to:

$$S_{score} = 0.5 + \frac{NT_{g1} + 2NT_{g2} + 0.02N_{g4} - NT_{g3b}}{23} - S_{aneuploidy} \quad (3)$$

where $NT_{g3b}$ is the number of trisomic chromosomes in group 3b and $NT_{g4}$ is the number of trisomic chromosomes in group 4. To save the computational memory requirements for exponential cell growth, virtual cells that died were removed from subsequent generations and 50,000 cells were randomly sampled into subsequent generations. If the number of virtual cells was less than 50,000, cells with aneuploid karyotypes were drawn from the pre-defined model and added to the current generation. Simulations were stopped when the UPID frequency of chromosomes 1–3, 5, 7–8, 11–13, 15, 16, 19, 20, and 22 became 2.5% or terminated after 2000 generations. Fifty parallel runs were performed for each model. After the end of the simulation, one million virtual cells were randomly sampled from each model and the karyotype similarity between the patient cohort and sampled cells was measured using the RMSE method.

To investigate whether the aneuploidy developed by punctuated or gradual evolution in the diploid/tripolar and diploid/sequential models, ten thousand virtual cells were randomly sampled from each simulated generation and the corresponding median modal chromosome number was calculated. One hundred parallel runs were performed and smoothing regression analysis (LOESS) was used to model the relationship between the modal chromosome number and the number of generations.

### Statistics and reproducibility
For assessing technical reproducibility, bulk WGS data from technical replicates represented by independent next generation sequencing libraries from the same DNA of 2 HeH samples (case L31 and case L74) were generated. A high correlation between the results from the two replicates was observed and over 97% of mutation sites were identified in the replication datasets. Since the reproducibility was very high, no additional replicates were generated. No statistical method was used to predetermine sample size. All cases with HeH where SNP array/WGS/WES data were available were included in the bulk copy number analysis, except for samples where the technical quality was too poor. The sister chromatid cohesion assay and the copy number variation calling were performed independently in a blinded fashion. All statistical tests were performed in R (version 4.1.2). The detailed statistical tests are indicated in figures or associated legends where applicable. No data were excluded from the analyses. None of the statistical tests used in this study required the assumption of normality or the assumption of equal variance. $P$ values were calculated based on nonparametric tests that do not have degrees of freedom associated with a sampling distribution. A significance threshold of <0.05 was used for all statistical tests.

### Reporting summary
Further information on research design is available in the Nature Portfolio Reporting Summary linked to this article.

## Data availability
The scWGS data generated in this study have been deposited in the European Genome Archive (EGA) under accession number EGAS00001006347. The scWGS dataset is available under restricted access due to privacy concerns; access can be obtained for academic research by contacting the Data Access Committee via EGA. The processed somatic SNP array data and bulk WGS data are freely available through the following DOIs: https://doi.org/10.17044/scilifelab.21953114 (SNP array dataset) and https://doi.org/10.17044/scilifelab.21953117 (bulk WGS dataset). The raw SNP array data and bulk WGS data generated during the current study have been deposited to EGA under accession numbers EGAS00001007049 and EGAS00001007052, respectively. These datasets are available under restricted access due to privacy concerns; access can be obtained for academic research by contacting the Data Access Committee via EGA. The WGS data generated by the Therapeutically Applicable Research to Generate Effective Treatments (TARGET) are available under accession code phs000464. The human reference GRCh37 (hg19) used in this study is available in the UCSC Genome Browser [http://hgdownload.cse.ucsc.edu/goldenPath/hg19/bigZips/]. Source data are provided with this paper.

## Code availability
The code used to perform the analysis is available as supplementary code, also available on Zenodo[55].

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

## Acknowledgements

The results published here are in part based upon data generated by the Therapeutically Applicable Research to Generate Effective Treatments (https://ocg.cancer.gov/programs/target) initiative, phs000218. Figures 1, 2, 4, and 5 and Supplementary Figs. 4, 7, and 8 were created with BioRender.com. This study was supported by grants from the Swedish Childhood Cancer Foundation, grant numbers PR2020-0033 (MY), TJ2020-0024 (MY), PR2018-0004 (BJ), and PR2018-0023 (KP); the Swedish Cancer Fund, grant numbers 20 0792 PjF (BJ) and 19-0252-Pj (KP); Governmental funding of clinical research within the National Health Service, grant number ALFSKANE-623431 (KP); the Swedish Research Council, grant numbers 2020-01164 (BJ) and 2020-00997 (KP); IngaBritt och Arne Lundbergs Forskningsstiftelse, grant number LU2019-0100 (KP), the Gunnar Nilsson Cancer Foundation (MY), the Royal Physiographic Society of Lund (EW), the Czech Health Research Council, grant number (NU20-07-00322)) (MZ)), the University Hospital Motol, grant number #00064203 (MZ, JZ), and Program EXCELES, grant number LX22NPO5102 (MZ, JZ).

## Author contributions

E.L.W., M.Y., and K.P. conceived the study; E.L.W., M.Y., L.H.M.-C., H.v.d.B., R.G., L.O.-A., and D.C.J.S. performed experiments and analyzed data; A.C., N.D., M.Z., J.Z., and B.J. provided clinical data and samples and analyzed data; F.F. supervised experiments and analyzed data; K.P. supervised the study; E.L.W., M.Y., and K.P. wrote the article with input from all authors.

## Funding

## Competing interests

The authors declare no competing interests.
