## [Peer Review File · Nature Communications]

REVIEWER COMMENTS

Reviewer #1 (Remarks to the Author): Expert in acute lymphoblastic leukaemia genomics, single-cell genomics, and evolution

Overall, this is a nice study, but mostly limited to correlations/associations, and no mechanistic insight is obtained; modelling provides some new insight, but remains to be confirmed. For example, the authors show that there is an association between gains/losses of chromosomes at disease relapse, but do not provide any evidence why this would be.

The only 'confirmation' comes from the simulations that are being performed. It remains unclear how good such simulations are, as there is no experimental validation of the simulated data/mechanisms.

The single-cell sequencing data is of high quality: the authors use low-pass single-cell whole genome sequencing of about 300 individual bone marrow cells/case, in total 2,847 cells, from nine primary hyperdiploid ALL cases.

The authors also use data from SNP arrays to study 577 additional hyperdiploid ALL cases. I believe this is indeed a good data set to identify the chromosomes that are most frequently affected, but I am not sure how sensitive these data are to determine subclones in the 577 cases: the authors claim that based on this data, they find that the majority of cases does not have subclones with other copy number changes. Can the authors give data to illustrate which subclones can be detected (what percentage of cells with different chromosomal changes than the major clone can be detected?). It will be required to validate this/describe this in more detail.

The modelling is of interest, but would need additional experimental validation to come to solid conclusions. For example, the authors conclude that "modelling in conjunction with the patient data suggest that, in most instances, high hyperdiploidy in pediatric ALL arises by a tripolar division in a diploid cell. However, HeH ALL with MCN 62-67 (comprising around 5% of cases) may possibly arise by initial tetraploidy followed by chromosomal losses."

It is correct that the authors use the words 'suggest' and 'may possibly arise by' but this indicates that these models are only suggesting and do not lead to solid conclusions.

Can the authors provide additional data to support the models and can the models be explained in more detail?

The observation that mutations occur subclonal and are therefore acquired later than the hyperdiploidy is not surprising and not completely novel.

Reviewer #2 (Remarks to the Author): Expert in cancer evolution, single-cell genomics, and bioinformatics

In this article, the authors used a combination of scWGS, SNP array and sequencing data from a large cohort of HeH ALL cases and showed that the chromosomal gains are early events and relatively stable throughout leukemia development. Through in silico simulation of five different models of high hyperdiploidy development, the authors also showed that tripolar division in a diploid cell, followed by clonal selection could be the most probable event underlying hyperdiploidy as this model best recapitulated the chromosomal patterns seen in patient samples. The authors also investigated BTRI and B/ATRI mutations in trisomic chromosomes and found that these mutations were associated with different mutational signatures. Overall, the article characterized the origin of aneuploidy in HeH ALL and suggested a punctuated evolution like-model. While most of the analyses presented are robust, I just have a few suggestions for the authors.

1. For copy number inference from scWGS, the authors have used the method Aneufinder. However, more recent and improved methods are available for copy number inference from single-cell data (e.g., <https://www.nature.com/articles/s41587-020-0661-6>). Can the authors run one of these new methods on their data and verify that the copy number calling by Aneufinder is indeed robust?
2. For inferring the phylogenetic tree, the authors used the dendrogram information from Aneufinder and manually curated scWGS, SNP array, and FISH results. This is an ad-hoc approach and it is not clear how different data sources from different patients can be integrated together for phylogeny inference. Phylogeny inference should be done for each patient (for scWGS data/WGS data) separately using a well-known computational approach (e.g., maximum parsimony has been used for single-cell copy number data). Specifically, for single-cell data, methods are already available (e.g., <https://genomebiology.biomedcentral.com/articles/10.1186/s13059-022-02693-z>).
3. The authors have suggested that aneuploidy arises by punctuated evolution at the early stages of tumorigenesis. Can the authors adopt an existing model of punctuated evolution and show that such a model fits the patient data well for HeH ALL? If such a comparison is out of scope, can the authors at least discuss how their hyperdiploidy simulation model aligns with a punctuated evolution model?

Reviewer #3 (Remarks to the Author): Expert in paediatric leukaemias and high hyperdiploid ALL

The paper by Woodward and colleagues demonstrates that the chromosomal gains characteristic of high hyperdiploid (HeH) ALL are an early event in transformation and gains occur simultaneously. While the findings largely validate previous work, lingering uncertainties are resolved. The scale of the analysis and the use of scWGS significantly advances the field. Moreover, the investigators were able to precisely describe phylogenetic trees and show positive and negative selection for certain trisomies during clonal evolution and at relapse. The work doesn't answer the essential question of what the underlying mechanisms of fitness are provided by individual trisomies but does provide a foundation to answer such questions.

Overall, the paper is extremely well written and should be of great interest to laboratory and clinical biologists. The conclusions are justified based on the extensive data presented.

Comments

It is interesting that chromosome 21 made up a majority of the 3:1 tetrasomies. Can the authors comment on why one homologue appeared to have a selective advantage?

It isn't entirely clear what the authors make of the mutational signature patterns (page 14). Are they suggesting that the early mutations before hyperdiploidy arises are background mutations in a stem cell and shared by normal cells (e.g., passengers)?

It is also interesting to note that many structural rearrangements, deletions and mutations in their cohort were subclonal and appeared to occur after the gain of chromosomes (page 15). Many of these lesions are known drivers of the leukemic process. Are they suggesting that in cases of HeH ALL they are less relevant? Given the long latency between simultaneous gains of chromosomes and overt leukemia what are the nature of second events that drive frank leukemogenesis?

Figure 3 might be better explained especially for those readers not quite as familiar with analytical pipelines (e.g., clinicians will find this article of great interest). In 3B the data is comparing results from the model to actual patient data. Thus the overlap is represented by percent on the Y axis, the higher the bar per each chromosome the more accurate the model is (diploid/tripolar)? But the relative

distribution of UPID, trisomy etc. (colors) is not clear. How does the distribution from the model overlap the distribution in patients? In general the distribution does appear to replicate what is in the narrative.

The authors indicate HeH cases with different MCN's might arise by different mechanisms. Is there anything in the data that might shed light on reasons for better clinical outcome in patients with the higher MCN?

Response to Reviewers

“Clonal origin and development of high hyperdiploidy in childhood acute lymphoblastic leukemia”

Woodward and Yang et al.

Considered for publication in Nature Communications

We thank the Reviewers for their constructive criticism that has significantly improved the manuscript. We have addressed their comments as detailed below and as marked by red text in the revised manuscript.

Reviewer #1.

Overall, this is a nice study, but mostly limited to correlations/associations, and no mechanistic insight is obtained; modelling provides some new insight, but remains to be confirmed. For example, the authors show that there is an association between gains/losses of chromosomes at disease relapse, but do not provide any evidence why this would be.

***Response:** That HeH ALL at diagnosis and relapse shows similar gains has been shown in several previous studies. To make this clearer, we have added the following sentence to the manuscript: “Such samples have previously been shown to be clonally related and display overall very similar karyotypes^{15,16}.” (page 9, section 2).*

The only ‘confirmation’ comes from the simulations that are being performed. It remains unclear how good such simulations are, as there is no experimental validation of the simulated data/mechanisms.

***Response:** Please see below for a detailed discussion on the important point of the validation of the simulated mechanisms.*

The single-cell sequencing data is of high quality: the authors use low-pass single-cell whole genome sequencing of about 300 individual bone marrow cells/case, in total 2,847 cells, from nine primary hyperdiploid ALL cases.

The authors also use data from SNP arrays to study 577 additional hyperdiploid ALL cases. I believe this is indeed a good data set to identify the chromosomes that are most frequently affected, but I am not sure how sensitive these data are to determine subclones in the 577 cases: the authors claim that based on this data, they find that the majority of cases does not have subclones with other copy number changes. Can the authors give data to illustrate which subclones can be detected (what percentage of cells with different chromosomal changes than the major clone can be detected ?). It will be required to validate this/describe this in more detail.

Response: This is a good point but unfortunately one that is not so straight-forward to address. Overall, the detection limit will depend a) on the purity of the leukemic sample, since contaminating normal blood cells will dilute the signal from the leukemia and making it harder to detect subclones and b) the type of subclonality. As regards a), this is not so much of a problem in HeH ALL, as the blast percentage is usually >90% in the bone marrow at the time of diagnosis, making these samples exceptionally pure in relation to other tumor types. Furthermore, as the blast percentage can easily be ascertained by bulk SNP array analysis by looking at B-allele frequency and log2 ratio, we estimate that <5% of the 577 HeH cases had blast percentages lower than 90%. As regards b), this is more complicated to address. We have made a dilution series of a diagnostic sample from a hyperdiploid leukemia with close to 100% leukemic cells and its corresponding remission sample (with 0% leukemic cells) and run this on Illumina Human 1M-Duo SNP array, to investigate at which frequency different types of aberrations are detectable. Using the software TAPS (which takes both B-allele frequency and log2 ratio into account and which we used in our manuscript), trisomies can be clearly detected at 30% leukemic cells, corresponding to disomy/trisomy subclonality being detectable at this frequency, whereas uniparental isodisomies can be detected at the level of 20%. These data have been added to the manuscript as Supplementary Figure 6.

In case 9 in the scWGS analysis, which had trisomy 17 in two different subclones corresponding together to 82% of the cells, it could clearly be seen in the bulk SNP array analysis that trisomy 17 was subclonal compared with the other trisomies.

TAPS results for chromosomes 17 (18% of cells disomic, 82% of cells trisomic), 18 (trisomic) and 19 (disomic) based on SNP array analysis results of case 9. Left: allelic imbalance vs. log2 ratio. Signals from the specific chromosome is shown in purple-red color. Signals cluster depending on copy number and allelic imbalance. In this particular case, clusters can be seen for disomies, uniparental isodisomies, trisomies, tetrasomies 2;2, and pentasomy. The signal from chromosome 17 falls between the trisomic and disomic clusters, showing subclonality. The subclonality is also visible in the B allele frequency on the right.

Thus, the limit of detection will differ depending on whether it is a major subclone with disomy or a major subclone with trisomy for disomy/trisomy subclonality, and also depending on the copy number in the two subclones (for example trisomy/UPID, trisomy/2:2 tetrasomy, trisomy/3:1 tetrasomy etc). As a general rule-of-thumb, though, the detection limit is approximately 20-30% for whole chromosome changes although it could be both lower and higher in certain scenarios.

With that in mind, the scWGS clearly shows that HeH ALL is generally chromosomally homogeneous. The purpose of analyzing subclonality in the 577 cases with bulk copy number data was not to investigate the frequency of subclones, but rather to use subclonality to gain insight into the selective pressures for individual chromosomes. For this, we focused on subclonality between 2 and 3 copies, the former either as a normal heterodisomy or as a uniparental isodisomy. The hypothetical detection limit (the shift in B allele frequency) for these is approximately the same in a hyperdiploid context where clonal trisomies are available for comparison. In this context, the overall ability of SNP array analysis to detect subclones is therefore less important. However, to make it clear to the reader that only relatively large subclones can be detected, we have added a sentence on this in the “Subclonality indicates selective pressures” section (page 8).

We also compared the frequency of subclonality detected by bulk copy number analysis in cases with 51-61 chromosomes and cases with 62-67 chromosomes, finding that subclonality of whole chromosomes was more common in the latter group. There should not be any difference in the detection limit depending on the modal number (the log2 ratio baseline will shift but the ability to detect subclonality will not be affected) and this comparison should therefore be valid.

The modelling is of interest, but would need additional experimental validation to come to solid conclusions. For example, the authors conclude that “modelling in conjunction with the patient data suggest that, in most instances, high hyperdiploidy in pediatric ALL arises by a tripolar division in a diploid cell. However, HeH ALL with MCN 62-67 (comprising around 5% of cases) may possibly arise by initial tetraploidy followed by chromosomal losses.” It is correct that the authors use the words ‘suggest’ and ‘may possibly arise by’ but this indicates that these models are only suggesting and do not lead to solid conclusions. Can the authors provide additional data to support the models and can the models be explained in more detail ?

Response: *We realize that we may have been too vague in the manuscript regarding the magnitude of the data that supports the tripolar division model as the main mechanism for HeH formation. In the absence of direct observations (which are not possible since the initial hit occurs in utero, one cell among millions), any proposed scenario for how hyperdiploidy arises can only be suggested, not proven. However, our scWGS experiments and deep analysis of the large patient cohort provide several observations and correlations that, as far as we can see, can only be explained by an initial tripolar division in a diploid cell.*

Specifically, any model for how high hyperdiploidy develops in ALL will have to explain the following observations and correlations:

- 1) The specific chromosomal pattern of trisomies, tetrasomies 2:2, a low (but nonetheless not zero) frequency of uniparental isodisomies, and no monosomies. This***

includes why 2:2 tetrasomies are so much more common than 3:1 tetrasomies and why uniparental disomies occur at a low but relatively constant frequency for different chromosomes (with the exception of chromosome 9). The tripolar division model recapitulates all of these features in the *in silico* analysis, in contrast to the other tested models.

- 2) **The overall (on a “population level”) presence of gains of all chromosomes at a low to very high frequency.** Specifically, any model should explain why the low frequency gains are there, as these are unlikely to be clonally selected for. In the tripolar division model, the low frequency gains arise as a side effect of the tripolar division and are in that sense passengers. Since they are not selected for, they will be gradually lost. This is also true for the other models except the consecutive gains in a diploid cell.
- 3) **The lack of overt chromosomal instability at diagnosis,** as shown by our scWGS and by e.g., observations of monozygotic twins with concurrent HeH ALL with identical/very similar karyotypes and of the similarities between diagnostic and relapse samples. This is in line with an origin via a tripolar division, with subsequent low level clonal evolution. Models including sequential gains/losses are also possible but would require very many cell divisions for a hyperdiploid karyotype to form.
- 4) **Common trisomies are on average newer than less common trisomies.** As we show in the manuscript, chromosomal gains that are selected for (the “strong_pos” and “weak_pos” groups) have more BTRI mutations, indicating that they on average occurred later, than chromosomal gains with neutral or negative selection (group “neg”). In a scenario where trisomies occur randomly throughout leukemogenesis (as in sequential gains), “driver” and “passenger” trisomies could occur at any order, as well as in scenarios starting with a tetraploid cell, where all would be expected to be equally old. However, in the tripolar division mechanism, all “passenger” trisomies (and many of the driver trisomies) arise in the first tripolar division whereas driver trisomies can also be gained later due to positive selection, explaining why the positively selected trisomies are newer.

To explain our reasoning more fully in the manuscript, we have substantially expanded the discussion on this issue (Discussion, pages 17-18). We hope that this will convince the Reviewer of the validity of our findings.

As regards cases with MCN 62-67, we have attempted to further validate our findings of a different (as opposed to cases with MCN 51-61) distribution of trisomies and tetrasomies using cytogenetic data from the Mitelman Database of Chromosome Aberrations and Gene Fusions in Cancer. Unfortunately, however, almost all cases had incomplete karyotypes (a common problem with HeH cases, which often have poor chromosome morphology), making this analysis impossible. Therefore, further proof of this mechanism will have to await new and larger patient cohorts becoming available. We do, however, consider our findings interesting and believe that they should be included in the manuscript, in particular as cases with higher modal chromosome numbers also are different clinically.

The observation that mutations occur subclonal and are therefore acquired later than the hyperdiploidy is not surprising and not completely novel.

Response: We agree that it is not novel that mutations are subclonal. The main purpose with including data on structural aberrations and SNVs in the manuscript was to give a complete picture of the genomic evolution in HeH ALL, utilizing the by far largest cohort of cases. However, to the best of our knowledge this is the first report including a systematic investigation showing that also (most) clonal mutations arise after the trisomies in HeH ALL.

Reviewer #2

In this article, the authors used a combination of scWGS, SNP array and sequencing data from a large cohort of HeH ALL cases and showed that the chromosomal gains are early events and relatively stable throughout leukemia development. Through in silico simulation of five different models of high hyperdiploidy development, the authors also showed that tripolar division in a diploid cell, followed by clonal selection could be the most probable event underlying hyperdiploidy as this model best recapitulated the chromosomal patterns seen in patient samples. The authors also investigated B/TRI and B/ATRI mutations in trisomic chromosomes and found that these mutations were associated with different mutational signatures. Overall, the article characterized the origin of aneuploidy in HeH ALL and suggested a punctuated evolution like-model. While most of the analyses presented are robust, I just have a few suggestions for the authors.

1. For copy number inference from scWGS, the authors have used the method Aneupfinder. However, more recent and improved methods are available for copy number inference from single-cell data (e.g., <https://www.nature.com/articles/s41587-020-0661-6>). Can the authors run one of these new methods on their data and verify that the copy number calling by Aneupfinder is indeed robust?

Response: To address this, we performed copy number calling with CHISEL (as suggested by the Reviewer) and compared the results with our aggregated data from Aneupfinder and Ginko (as reported in the manuscript). CHISEL yielded very clear normalized read depth ratio plots. However, the copy number estimation using phased germline SNPs frequently changed a correct “HAP_CN” call to an incorrect “CORRECTED_HAP_CN” call, as evidenced clearly in the read depth ratio plot. Below is an example, where, based on the estimated read-depth ratio (RDR) plot and the raw total copy number calling result (HAP_CN), chr1:220Mb-240Mb of the following cell should had 3 copies, while “CORRECTED_HAP_CN” called 2 copies. We believe these errors resulted from the ultra low-pass whole genome sequencing method (0.02x) that we used, leading to an insufficient number of informative heterozygous variants in the scWGS dataset for the algorithm to work properly on our data. Thus, we deem CHISEL unsuitable for the analysis of our scWGS cases.

Estimated read depth ratio (RDR) plot from CHISEL of a cell from case 2, clearly showing copy number gain for the whole 1q.

CHISEL copy number segmentation result for the same cell from case 2.

#CHR	START	END	CELL	NORM_COUNT	COUNT	RDR	A_COUNT	B_COUNT	BAF	CLUSTER	HAP_CN	CORRECTED_HAP_CN
chr1	0	10000000	CCGGTGCTTAAG	2341	853	0.849583774	19	11	0.366666667	27	1 1	1 1
chr1	10000000	20000000	CCGGTGCTTAAG	2223	758	0.833250182	6	18	0.75	32	1 1	1 1
chr1	20000000	30000000	CCGGTGCTTAAG	2461	886	0.891084033	3	14	0.823529412	49	1 1	1 1
chr1	30000000	40000000	CCGGTGCTTAAG	2474	821	0.823293305	9	3	0.25	1	1 1	1 1
chr1	40000000	50000000	CCGGTGCTTAAG	2471	899	0.919047067	6	7	0.538461538	1	1 1	1 1
chr1	50000000	60000000	CCGGTGCTTAAG	2514	758	0.799879554	9	3	0.25	37	0 2	1 1
chr1	60000000	70000000	CCGGTGCTTAAG	2475	691	0.780483228	10	0	0	4	2 0	2 0
chr1	70000000	80000000	CCGGTGCTTAAG	2548	709	0.816729088	10	0	0	22	2 0	2 0
chr1	80000000	90000000	CCGGTGCTTAAG	2422	646	0.770637508	10	0	0	4	2 0	2 0
chr1	90000000	100000000	CCGGTGCTTAAG	2488	751	0.856002499	10	0	0	4	2 0	2 0
chr1	100000000	110000000	CCGGTGCTTAAG	2336	611	0.766862773	10	0	0	4	2 0	2 0
chr1	110000000	120000000	CCGGTGCTTAAG	2435	760	0.823804573	5	4	0.444444444	49	1 1	1 1
chr1	150000000	160000000	CCGGTGCTTAAG	2514	1249	1.263873457	6	7	0.538461538	9	1 2	2 1
chr1	160000000	170000000	CCGGTGCTTAAG	2398	1105	1.238773573	10	0	0	5	3 0	3 0
chr1	170000000	180000000	CCGGTGCTTAAG	2408	1103	1.270362224	10	0	0	5	3 0	3 0
chr1	180000000	190000000	CCGGTGCTTAAG	2487	1112	1.26448639	10	0	0	5	3 0	3 0
chr1	190000000	200000000	CCGGTGCTTAAG	2531	987	1.176197404	10	0	0	25	3 0	3 0
chr1	200000000	210000000	CCGGTGCTTAAG	2399	1278	1.349258378	13	16	0.551724138	34	1 2	2 1
chr1	210000000	220000000	CCGGTGCTTAAG	2471	1133	1.277314446	10	0	0	5	3 0	3 0
chr1	220000000	230000000	CCGGTGCTTAAG	2446	1121	1.194996895	6	13	0.684210526	42	2 1	1 1
chr1	230000000	240000000	CCGGTGCTTAAG	2507	1155	1.251676022	8	11	0.578947368	42	1 2	1 1
chr1	240000000	249250621	CCGGTGCTTAAG	2112	1031	1.339054716	6	4	0.4	38	1 2	1 1

We instead continued by trying SCOPE; a newly published software for copy number calling for scWGS data (Wang et al., Cell Systems 2020;10:445). SCOPE showed extremely high overlap with the curated data based on Aneupfinder and Ginko in our manuscript, with only 11 (0.38%) out of the total of 2,842 cells showing discrepant results, and in those only the call for one particular copy number segment. As any copy number callers are likely to give rise to small

differences in the calling, and in light of the overall very good agreement between our initial copy number calling and SCOPE, we believe that it is reasonable to keep the results for the scWGS copy number as they are in the manuscript.

Comparison between data in manuscript and copy number calls made by SCOPE

Case	No of cells	No of cells showing copy number calls in complete agreement	No of cells showing discrepant copy number calls (%)
1	269	268	1 (0.37%)
2	257	253	4 (1.6%)
3	272	270	2 (0.74%)
4	348	348	0 (0%)
5	347	345	2 (0.58%)
6	271	271	0 (0%)
7	273	273	0 (0%)
8	266	264	2 (0.75%)
9	269	269	0 (0%)
NBM	270	270	0 (0%)
Total	2842	2835	11 (0.38%)

2. For inferring the phylogenetic tree, the authors used the dendrogram information from Aneupfinder and manually curated scWGS, SNP array, and FISH results. This is an ad-hoc approach and it is not clear how different data sources from different patients can be integrated together for phylogeny inference. Phylogeny inference should be done for each patient (for scWGS data/WGS data) separately using a well-known computational approach (e.g., maximum parsimony has been used for single-cell copy number data). Specifically, for single-cell data, methods are already available (e.g., <https://genomebiology.biomedcentral.com/articles/10.1186/s13059-022-02693-z>).

Response: *In regard to how different data sources were integrated, this was done for each individual case/tree, not including data from different patients. Bulk SNP array/WGS was used to curate the trees regarding breakpoints leading to copy number changes and homologue inheritance. FISH was done to clarify how copy number changes involving parts of chromosomes corresponded to structural changes in cases 3 and 4 (as shown in Supplementary Figure 4). This was necessary to achieve correct phylogeny for structural rearrangements, since these cannot arise and/or clonally evolve independently (e.g., a derivative chromosome being lost would have to lead to loss of segments from both involved chromosomes, not only one of them) in the way that a simpler copy number change (e.g. a deletion) could.*

To address the Reviewer’s concern, we have now performed phylogeny inference with the novel MEDDIC2 software, developed for analysis of copy number changes from scWGS (Kaufmann et al., MEDICC2: whole-genome doubling aware copy-number phylogenies for cancer evolution. doi: <https://doi.org/10.1101/2021.02.28.433227>). Overall, the results showed good agreement with the trees presented in the manuscript, please see below for a detailed description of each case and when applicable the rationale for adjusting the trees. To root the trees, we used the presence of structural aberrations in subclones, as these are highly unlikely to disappear once established. We have updated the trees for cases 3, 4, 7 and 8 in the revised

version of the manuscript (also leading to minor changes in the Results section on page 6) and added MEDDIC2 in the Methods. This did not change the overall conclusions drawn from the phylogenetic analysis.

Detailed analysis of phylogenetic trees for cases 1-9

Case 1

The phylogenetic tree generated by the MEDIC2 software is identical to the one in our manuscript. We can infer that clone C is most likely to be the root of the tree. For any other clone to be the root, loss of structural aberrations would have to occur, which is unlikely.

MEDIC2 Phylogenetic Tree Case 1:

Case 2

From the phylogenetic tree generated by the MEDIC2 software, we can infer that the clone most likely to be the root of the tree is clone G. For any other clone to be the root, loss of structural aberrations would have to occur, which is unlikely. Whilst the lower half of the MEDIC2-generated tree is in agreement with the tree in our manuscript, the progression of clone F to D to I is simply not plausible as this would require sequential gain, loss, and then gain again of chromosome 21 over the course of clonal evolution. We also think that the progression of clone F to clone H is unlikely to occur. Gains of 1q (*dup(1q)*) in ALL usually arises either by internal tandem duplications or unbalanced translocations. Although they may be lost or gained over the course of clonal evolution (e.g. between diagnosis and relapse), changes in their structure, once formed, have not been reported in the literature. Thus, we deem

it unlikely for clone H to lose part of the 1q previously gained. We argue that our original hypothesized phylogenetic tree is the most likely route of clonal evolution.

MEDICC2 Phylogenetic Tree Case 2:

Case 3

This is one of the most genomically complex cases. By FISH, we have found that the copy number gains arise from two different structural rearrangements involving chromosomes 1 and 16 (see Supplementary Figure 4), with breakpoints that we could track in the bulk WGS data. These structural rearrangements lead to different combinations of copy number changes and can therefore be tracked in the scWGS data. The tree in our manuscript agrees quite well with that generated from the software. However, it is unclear to us why clones A, B, and J are set to originate from an “unknown” clone, as clones B and J both contain the *der(16)t(16;16),der(?)t(?;1)* aberration (i.e., identical combinations of copy number changes in chromosomes 1 and 16) seen in clone A as well as additional structural aberrations, suggesting they both originate from clone A. We also disagree with the placement of clones A, C, and D. Clones A and C have different combinations of copy number changes for chromosomes 1 and 16 corresponding to two independent structural rearrangements (Supplementary Figure 4). Thus, they should be in separate parts of the tree and not portrayed as one clone arising from the other. It may be possible for clone E to originate from clone C instead of the other way around, but we cannot be certain of the direction of evolution. The positioning of clone K is problematic, with inconsistencies arising from its placement on either side of the tree. We have updated the phylogenetic tree in the manuscript to the software’s positioning of clones C, E and K (Figure 2).

MEDICC2 Phylogenetic Tree Case 3:

Case 4

This is also one of the more complex cases genomically. Based on FISH, it has a structural rearrangement involving chromosome 14, leading to an add(14)q12, that is present in one or two copies (with 3 and 4 normal chromosome 14 homologues, respectively) (Supplementary Figure 4). Based on the MEDICC2 phylogenetic tree, it is possible that in this case the root clone was a pentasomy X that evolved into clones E and H/J to create two separate branches. Regarding the course of evolution from clone E, we feel that neither two separate events of a 9q gain with identical breakpoints nor two separate events of extra add(14)(q12) with loss of a normal chromosome 14 are plausible. However, applying the principle that it is more unlikely for a structural rearrangement with the same breakpoint to occur twice than a copy number change, we agree that the suggested evolution of clone D from B would be more likely to occur. In the other part of the tree, it is extremely unlikely that clone J emerged from clones E, H, and I as this would have involved sequential gains of the p- and q-arms of chromosome 6 (with the same breakpoint) over the course of the clonal evolution. It is far more likely that a whole chromosome 6 gain occurred once in a common predecessor to clones H and J. Taken together, we have updated the tree in the manuscript regarding clones A, B, D and E.

MEDICC2 Phylogenetic Tree Case 4:

Case 5

The phylogenetic tree generated by the MEDICC2 software is identical to the one in our manuscript. We can infer that the clone most likely to be the root of the tree is clone A. For any other clone to be the root loss of structural aberrations would have to occur, which would be unlikely.

MEDICC2 Phylogenetic Tree Case 5:

Case 6

The phylogenetic tree generated by the MEDICC2 software is identical to the one in our manuscript. We can infer that the clone most likely to be the root of the tree is clone A. For any other clone to be the root loss of structural aberrations would have to occur, which would be unlikely.

MEDICC2 Phylogenetic Tree Case 6:

*Common karyotype: +X,+4,+6,+9,+10,+18,+21,+21

Case 7

The phylogenetic tree generated by the MEDICC2 software is identical to the one in our manuscript. It is possible for either clone A or clone B to be the root of the tree as the direction of chromosomal change is not clear. We have added this to the tree in the manuscript.

MEDICC2 Phylogenetic Tree Case 7:

*Common karyotype: +X,+4,+6,+14,+17,+18,+21,+21

Case 8

The phylogenetic tree generated by the MEDICC2 software is identical to the one in our manuscript. It is possible for either clone A or clone C to be the root of the tree as the direction of chromosomal change is not clear. We have added this to the tree in the manuscript.

MEDICC2 Phylogenetic Tree Case 8:

*Common karyotype: +X,+4,+14,+21,+21

Case 9

The phylogenetic tree generated by the MEDICC2 software is identical to the one in our manuscript. The root of this tree has to begin with a disomy 17 due to the parental inheritance pattern of the trisomy 17 observed in clones A and B (and their daughter clones). We can infer that the clone most likely to be the root of the tree is clone F, since clones G or H being the root would involve structural changes to be lost, which is unlikely.

MEDICC2 Phylogenetic Tree Case 9:

*Common karyotype: +X,+4,+4,+8,+14,+18,+21,+21,+21

3. The authors have suggested that aneuploidy arises by punctuated evolution at the early stages of tumorigenesis. Can the authors adopt an existing model of punctuated evolution and show that such a model fits the patient data well for HeH ALL? If such a comparison is out of scope, can the authors at least discuss how their hyperdiploidy simulation model aligns with a punctuated evolution model?

Response: *To address this, we have now performed phylogenetic analyses using the Analysis of Phylogenetics and Evolution (ape) software (Paradis and Schliep, Bioinformatics 2019;35:526), which provides analysis of evolutionary distances (diploid cell to most recent common ancestor (MRCA) and MRCA to the terminal node, for the nine cases investigated with scWGS. The resulting trees fit very well with punctuated copy number evolution in all cases, with long truncal distances and short branching distances. These data have been added to the manuscript (Results page 3, new Supplementary Figure 3, and Methods page 20). We also performed sampling of the karyotypes in the in silico analysis to show that the average chromosome modal number distribution in the diploid/tripolar model agreed with punctuated evolution, in contrast to the diploid/sequential model (Results page 12 and Methods page 26). Finally, we have added a section in the Discussion on punctuated evolution in relation to our data (page 18).*

Reviewer #3

The paper by Woodward and colleagues demonstrates that the chromosomal gains characteristic of high hyperdiploid (HeH) ALL are an early event in transformation and gains occur simultaneously. While the findings largely validate previous work, lingering uncertainties are resolved. The scale of the analysis and the use of scWGS significantly advances the field. Moreover, the investigators were able to precisely describe phylogenetic trees and show positive and negative selection for certain trisomies during clonal evolution and at relapse. The work doesn't answer the essential question of what the underlying mechanisms of fitness are provided by individual trisomies but does provide a foundation to answer such questions.

Overall, the paper is extremely well written and should be of great interest to laboratory and clinical biologists. The conclusions are justified based on the extensive data presented.

Comments

It is interesting that chromosome 21 made up a majority of the 3:1 tetrasomies. Can the authors comment on why one homologue appeared to have a selective advantage?

Response: *We believe that chromosome 21 is overrepresented in the 3:1 tetrasomies due to the very high positive selection pressure for this chromosome, not because one homologue has a selective advantage. For all cases that have a trisomy that becomes a tetrasomy, the likelihood of that tetrasomy being 3:1 is 2/3 and it being 2:2 is 1/3 because one homologue is already present in two copies. Since the selective pressures are strongest for extra gains of chromosome 21 (as shown in our analysis), there will be an enrichment for 3:1 tetrasomy 21, as also corroborated by the in silico modelling, where the selective advantage was the same for both*

chromosome 21 homologues and still 3:1 tetrasomies accumulated. To make this clearer, we have added “Notably, 3:1 tetrasomies were not an indication of one homologue being selected for, but rather resulted from the strong overall selection for extra copies of chromosome 21 in both models.” in the first section on page 12.

When checking these data, we also discovered an error in the given frequencies of 3:1 tetrasomy 21 in patient samples, diploid/sequential, and diploid/tripolar models in the first section on page 12. The frequencies shown in Figure 3B were correct though, and the conclusions did not change. We have now corrected the given frequencies in the text and apologize for this error.

It isn't entirely clear what the authors make of the mutational signature patterns (page 14). Are they suggesting that the early mutations before hyperdiploidy arises are background mutations in a stem cell and shared by normal cells (e.g., passengers)?

Response: *The mutational signatures are not in themselves indicative of mutations being passengers (or drivers). Our main purpose with separating BTRI and B/ATRI mutations in this analysis was to show that BTRI mutations, and thereby the hyperdiploidy, arise very early in life, in line with a long latency after the initial hit (according to our results a tripolar division). However, it is likely that the vast majority of mutations (both ATRI and B/ATRI) are passengers, in particular as most occur in non-coding regions of the genome.*

It is also interesting to note that many structural rearrangements, deletions and mutations in their cohort were subclonal and appeared to occur after the gain of chromosomes (page 15). Many of these lesions are known drivers of the leukemic process. Are they suggesting that in cases of HeH ALL they are less relevant? Given the long latency between simultaneous gains of chromosomes and overt leukemia what are the nature of second events that drive frank leukemogenesis?

Response: *We agree that this is an interesting finding. We believe that these lesions are still as important in the leukemogenesis of HeH ALL as in other genetic subtypes even though they occur at a later stage of leukemogenesis. As the reviewer states, some of them may even be necessary for full blown leukemia, although this is beyond the scope of our study to investigate.*

Figure 3 might be better explained especially for those readers not quite as familiar with analytical pipelines (e.g., clinicians will find this article of great interest). In 3B the data is comparing results from the model to actual patient data. Thus the overlap is represented by percent on the Y axis, the higher the bar per each chromosome the more accurate the model is (diploid/tripolar)? But the relative distribution of UPID, trisomy etc. (colors) is not clear. How does the distribution from the model overlap the distribution in patients? In general the distribution does appear to replicate what is in the narrative.

Response: *We apologize for the, in hindsight, insufficient figure legend. We have substantially expanded the figure legend and hope that this will clarify the results. Figure 3B shows the end results of the simulations, i.e. the resulting chromosomal pattern, so that they can be visually*

compared with the pattern in the patient cohort. The results from the statistical test, investigating the similarity between the diploid/tripolar model and the patient cohort is shown in Supplementary Table 5.

The authors indicate HeH cases with different MCN's might arise by different mechanisms. Is there anything in the data that might shed light on reasons for better clinical outcome in patients with the higher MCN?

Response: *Unfortunately, we did not find any explanation for the better clinical outcome in HeH patients with higher MCN, something which is indeed a very important issue in light of the clinical implications. However, that we see a possible different distribution of trisomies and tetrasomies (as compared with cases with lower MCN) in the MCN 62-67 group supports that these may indeed be biologically different. We hope to address this again in the future when larger patient cohorts become available.*

REVIEWERS' COMMENTS

Reviewer #2 (Remarks to the Author):

I sincerely thank the authors for performing additional analyses to address my concerns. I think the new analyses have improved the manuscript and it is ready for publication.

Reviewer #3 (Remarks to the Author):

The manuscript by Woodward and colleagues describes their studies into the genetic mechanisms leading to high hyperdiploid B acute lymphoblastic leukemia, the most common biological subtype. While their work is consistent with previous studies, the level of detail provided by single cell whole genome sequencing on a large cohort of cells clears up any lingering uncertainties. Their results support a single tripolar mitosis as the initiating event over competing models. Moreover, they report additional data showing selective pressures at the clonal and subclonal level for certain chromosomes. The data set will serve as a foundation for further studies examining the biological basis for non-random chromosome gains and the favorable outcome associated with this subgroup.

The authors have responded to the constructive comments provided in the initial review and parts of the narrative have been edited for clarity.

Reviewer #4 (Remarks to the Author):

I am a new Reviewer, replacing Reviewer 1, and have been asked by the Editor to limit my evaluation to the response to the original Reviewer. I think the authors have largely addressed the concerns raised by Reviewer 1. The main sticking point in my view is the response to the question the Reviewer raised about the sensitivity of subclone calling. The response is not satisfactory.

What the Reviewer is asking is: How good is the sensitivity for calling subclones? The answer requires a comparison against some sort of gold standard which really is bulk WGS data, not SNP arrays. So what I would suggest is this: take 3 samples with varying purity (high, middle, low) and perform deep WGS (200X ~ corresponding to the Illumina error rate) and then determine the clonal structure based on CN

changes. In doing so, the authors ought to use methods that are designed for subclonal CN calling (eg, Battenberg). They could further improve their sensitivity by using the definitive biological phasing of SNPs which can be defined for chromosomes with allelic imbalances. This will provide gold standard readouts for the definition of subclones against which they can benchmark their approach. This WGS experiment will also assess clonal diversity based on substitutions; there may be extensive diversification hidden.

Second response to Reviewers

“Clonal origin and development of high hyperdiploidy in childhood acute lymphoblastic leukemia”

Woodward and Yang et al.

Considered for publication in Nature Communications

Again, we thank the Reviewers for their constructive criticism that has significantly improved the manuscript. We have addressed their comments as detailed below and as marked by red text in the revised manuscript.

Reviewer #2 (Remarks to the Author):

I sincerely thank the authors for performing additional analyses to address my concerns. I think the new analyses have improved the manuscript and it is ready for publication.

Response: We were pleased to read that the Reviewer is now satisfied with our revised manuscript and thank the Reviewer for constructive criticism that improved the manuscript.

Reviewer #3 (Remarks to the Author):

The manuscript by Woodward and colleagues describes their studies into the genetic mechanisms leading to high hyperdiploid B acute lymphoblastic leukemia, the most common biological subtype. While their work is consistent with previous studies, the level of detail provided by single cell whole genome sequencing on a large cohort of cells clears up any lingering uncertainties. Their results support a single tripolar mitosis as the initiating event over competing models. Moreover, they report additional data showing selective pressures at the clonal and subclonal level for certain chromosomes. The data set will serve as a foundation for further studies examining the biological basis for non-random chromosome gains and the favorable outcome associated with this subgroup.

The authors have responded to the constructive comments provided in the initial review and parts of the narrative have been edited for clarity.

Response: We were pleased to read that the Reviewer is now satisfied with our revised manuscript and thank the Reviewer for constructive criticism that improved the manuscript.

Reviewer #4 (Remarks to the Author):

I am a new Reviewer, replacing Reviewer 1, and have been asked by the Editor to limit my evaluation to the response to the original Reviewer. I think the authors have largely addressed the concerns raised by Reviewer 1. The main sticking point in my view is the response to the question the Reviewer raised about the sensitivity of subclone calling. The response is not satisfactory.

What the Reviewer is asking is: How good is the sensitivity for calling subclones? The answer requires a comparison against some sort of gold standard which really is bulk WGS data, not SNP arrays. So what I would suggest is this: take 3 samples with varying purity (high, middle, low) and perform deep WGS (200X ~ corresponding to the Illumina error rate) and then determine the clonal structure based on CN changes. In doing so, the authors ought to use methods that are designed for subclonal CN calling (eg, Battenberg). They could further improve their sensitivity by using the definitive biological phasing of SNPs which can be defined for chromosomes with allelic imbalances. This will provide gold standard readouts for the definition of subclones against which they can benchmark their approach. This WGS experiment will also assess clonal diversity based on substitutions; there may be extensive diversification hidden.

Response: *Firstly, we would like to thank the Reviewer for performing this review of our response to Reviewer 1.*

Regarding the remaining issue of the sensitivity of SNP arrays to detect copy number subclonality, we believe that this question from Reviewer 1 could be due to misunderstanding our purpose with analyzing subclonality with SNP array analysis. Our main purpose with analyzing subclonality in the SNP array data was to investigate the direction of selection for individual chromosomal gains, not to investigate the level of subclonality as such, which would require additional experiments as suggested by Reviewer 4.

To investigate the direction of selection, the absolute size of the subclones is not important as long as the detection limit for subclonality is approximately the same for the types of subclonality that are compared (in this case uniparental isodisomy/trisomy and normal heterodisomy/trisomy). As we wrote in our response to Reviewer 1, the expected shift in B-allele frequency is approximately the same for these two types of subclonality, if clonal trisomies are present for comparison (which they always are in HeH cases). Thus, we believe that further investigations into the exact detection limits for whole chromosome gains with SNP array would be beside the point and not add to or change the results of the present study.

We absolutely agree with Reviewers 1 and 4 that if bulk samples were used to discover the true percentage of different subclones and to detect low-level subclones, deep WGS would be more suited than SNP array analysis and that Reviewer 4's suggestion would be useful to determine the limits of the latter method. However, we believe that neither method is very good for detecting subclonality involving copy number changes in small subclones. For that, scWGS is clearly superior as the true phylogeny can be assessed including gains of the same chromosome in different branches of the tree, as we indeed show in our investigation.

Finally, we fully agree with Reviewer 4 that deep WGS would be valuable in revealing subclonality associated with SNVs, something that has still not been thoroughly investigated in HeH ALL. However, although interesting, this is beyond the scope of the current investigation.

To make the purpose and limitations of using SNP array analysis to study subclones more clear, we have rewritten the section “Subclonality indicates selective pressures” on pages 8-9 extensively.